# NEUROPROBE: EVALUATING INTRACRANIAL BRAIN RESPONSES TO NATURALISTIC STIMULI

## ABSTRACT

High-resolution neural datasets enable foundation models for the next generation of brain-computer interfaces and neurological treatments. The community requires rigorous benchmarks to discriminate between competing modeling approaches, yet no standardized evaluation frameworks exist for intracranial EEG (iEEG) recordings. To address this gap, we present Neuroprobe: a suite of decoding tasks for studying multi-modal language processing in the brain. Unlike scalp EEG, *intracranial* EEG requires invasive surgery to implant electrodes that record neural activity directly from the brain with minimal signal distortion. Neuroprobe is built on the BrainTreebank dataset, which consists of 40 hours of iEEG recordings from 10 human subjects performing a naturalistic movie viewing task. Neuroprobe serves two critical functions. First, it is a mine from which neuroscience insights can be drawn. The high temporal and spatial resolution of the labeled iEEG allows researchers to systematically determine when and where computations for each aspect of language processing occur in the brain by measuring the decodability of each feature across time and all electrode locations. Using Neuroprobe, we visualize how information flows from key language and audio processing sites in the superior temporal gyrus to sites in the prefrontal cortex. We also demonstrate the progression from processing simple auditory features (e.g., pitch and volume) to more complex language features (part of speech and word position in the sentence tree) in a purely data-driven manner. Second, as the field moves toward neural foundation models trained on large-scale datasets, Neuroprobe provides a rigorous framework for comparing competing architectures and training protocols. We found that the linear baseline on spectrogram inputs is surprisingly strong, beating frontier foundation models on many tasks. Neuroprobe is designed with computational efficiency and ease of use in mind. We make the code for Neuroprobe openly available and will maintain a public leaderboard of evaluation submissions, aiming to enable measurable progress in the field of iEEG foundation models.

## 1 INTRODUCTION

The human brain constantly engages in a variety of simultaneous processing tasks: parsing speech, interpreting dynamic visual scenes, performing social reasoning, and integrating multi-modal sensory information (Schurz et al., 2014). However, our understanding of how this processing is organized across time and brain regions remains incomplete, and decoding the contents of these computations in the brain remains a difficult task (Paninski & Cunningham, 2018). A central challenge is that traditional approaches have been limited by small-scale datasets and simplified experimental paradigms that isolate individual tasks (Nastase et al., 2020), rather than study tasks concurrently.

Recent advances in data collection have created new opportunities to address these limitations through large-scale human intracranial electroencephalography (iEEG) datasets (Peterson et al., 2022; Evanson et al., 2025; Zada et al., 2025; Wang et al., 2024). These datasets, collected from neurosurgical patients undergoing clinical monitoring, approach the data volumes that have enabled breakthroughs in other domains of machine learning. Intracranial EEG differs substantially from scalp EEG. While scalp EEG suffers from significant signal distortion as neural activity passes through the skull, cerebrospinal fluid, and scalp tissues (Nunez & Srinivasan, 2006), iEEG electrodes record directly from the brain surface or within brain tissue, offering a substantially higher-

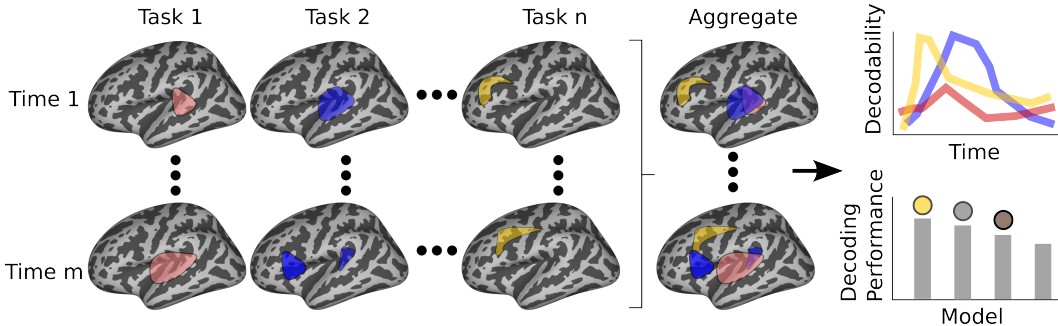

Figure 1: **Overview of Neuroprobe's goals.** Neuroprobe consists of classification tasks derived from human intracranial recordings aligned with annotated stimuli. It serves two critical roles: first, by performing a decoding analysis for each task, we can localize various aspects of multimodal language processing in the brain and discover their time evolution. Second, Neuroprobe is a rigorous, standardized benchmark for evaluating neural decoding models, which fills a critical need for the iEEG foundation model community.

fidelity signal. For example, intracranial EEG preserves high-frequency bands (e.g., high-gamma activity above 70 Hz) that are largely lost in scalp EEG due to filtering and noise (Ray & Maunsell, 2011; Lachaux et al., 2012). These high-frequency signals are closely linked to local computation and population spiking, making intracranial recordings essential for many decoding tasks.

The emergence of foundation models of neural activity offers new possibilities for leveraging these large-scale iEEG datasets. Recent developments such as Neuroformer (Antoniades et al., 2024), BrainBERT (Wang et al., 2023b), PopT (Chau et al., 2024), STNDT (Le & Shlizerman, 2022), NDT2 (Ye et al., 2023), MBrain (Cai et al., 2023), Brant (Zhang et al., 2023), MtM (Zhang et al., 2024b), and POYO (Azabou et al., 2023), and others demonstrate the potential for self-supervised learning approaches to extract meaningful representations from neural data. These foundation models achieve superior decoding performance across multiple tasks, which directly translates to increased statistical power for experiments that identify when and where specific cognitive processes occur in the brain. Similar probing experiments have been previously used successfully in the field of machine learning interpretability to reverse engineer neural networks by identifying where certain features of stimuli first become decodable (Tenney et al., 2019; Alain & Bengio, 2016). Performant iEEG foundation models have the potential to unlock novel insights about the brain, as well as enable the next generation of brain-computer interfaces and neurological treatments.

However, the iEEG community currently lacks the standardized evaluation frameworks necessary to rigorously compare these emerging approaches. For example, a recent review by Kuruppu et al. (2025) identifies this lack of common standardized evaluation and stresses that establishing a common benchmark is essential for comparing the performance of EEG foundation models performance and measuring advances in the field.

To address these critical gaps, we introduce *Neuroprobe*, a benchmark that is designed both to be a setting in which neuroscience probing experiment may be run *and* as a measure of progress in the field of iEEG foundation models (Figure 1). Our benchmark is derived from the publicly available BrainTreebank dataset (Wang et al., 2024), which consists of intracranial neural recordings aligned with the corresponding movie stimuli. Neuroprobe turns this dataset into a benchmark by defining 15 decoding tasks that span the audio, vision and language domains.

We have designed Neuroprobe to be computationally efficient and convenient for use by members of the machine learning community, even if they do not have deep domain expertise in neuroscience. By lowering the barrier of entry, we hope to create a healthy community and attract more researchers to these important problems. We standardize a number of aspects of the benchmark. We select train/test splits in a variety conditions: from training and testing on the same subject and session, to doing cross-subject and cross-session decoding. Finally, we host a centralized website that aggregates results, and displays a leaderboard that tracks progress in decoding performance of iEEG foundation models.

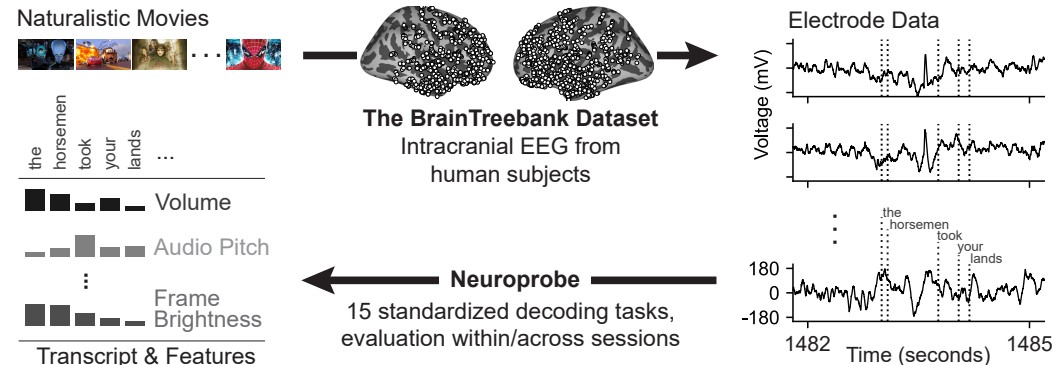

Figure 2: **From raw data to decoding tasks.** As part of the BrainTreebank dataset, 26 movies (left) are watched by 10 patients with stereoelectroencephalography intracranial electrodes implanted in various brain regions (middle), and the local field potential from the implanted electrodes is recorded (right). Neuroprobe turns this dataset into a standardized evaluation benchmark by segmenting the aligned data into various audio, language, and vision decoding tasks, such as volume, pitch, average frame brightness, etc.

In summary:

1. We introduce Neuroprobe, a large-scale multitask decoding benchmark for human intracranial EEG.
2. We standardize splits and metrics to rigorously evaluate iEEG foundation models and encourage their development in a direction which benefits decoding across many tasks.
3. We establish a set of strong baselines and compute the performance of state-of-the-art models on Neuroprobe.
4. Using Neuroprobe, we visualize the spatial distribution of different task processing pathways in the brain, and track their evolution across time.

In the long run, we aim for Neuroprobe to enable measurable progress in the field of iEEG foundation models, and lead to an improved understanding of the computations behind multi-modal sensory processing in the brain.

## 2 RELATED WORK

**Neural recording datasets**   The most recently developed models for neural data have relied on several widely accessible datasets. For non-invasive scalp EEG decoding, datasets from Zheng & Lu (2015); Grootswagers et al. (2022); Bhattasali et al. (2020); Tangermann et al. (2012); Obeid & Picone (2016); Broderick et al. (2018); Brennan & Hale (2019) have been used in the construction of models such as those proposed by Jiang et al. (2024); Yang et al. (2023); Défossez et al. (2023). For fMRI decoding, (Wehbe et al., 2014; LeBel et al., 2023; Nastase et al., 2021; Li et al., 2022; Allen et al., 2022) have led to models such as those proposed by Scotti et al. (2024); Ozcelik & VanRullen (2023). For MEG decoding, Jan-Mathijs et al. (2019); Hebart et al. (2023) released data that have supported training of models such as those proposed by Défossez et al. (2023); Benchetrit et al.. For neural spike decoding, data by Perich et al. (2025); Churchland et al. (2024); Manley et al. (2024); IBL (2024) enabled foundation models such as POYO and NDT (Azabou et al., 2023; Zhang et al., 2024a). Finally, for broadband intracranial neural activity, datasets from (Peterson et al., 2022; Wang et al., 2024; Nejedly et al., 2020) have fueled the development of iEEG foundation models proposed by Peterson et al. (2021); Wang et al. (2023b); Chau et al. (2024); Yuan et al. (2024); Zhang et al. (2023). However, these datasets do not provide rigorous splits or testing guidelines, so each model is difficult to compare to others. Other ECOG datasets include Chen et al. (2024); Zheng et al. (2024); Singh et al. (2025).

**Existing neural data benchmarks**   In the field of machine learning for neuroscience, benchmarks exist across various neural data modalities. Some of the earliest involve EEG BCI decoding (Tanger-

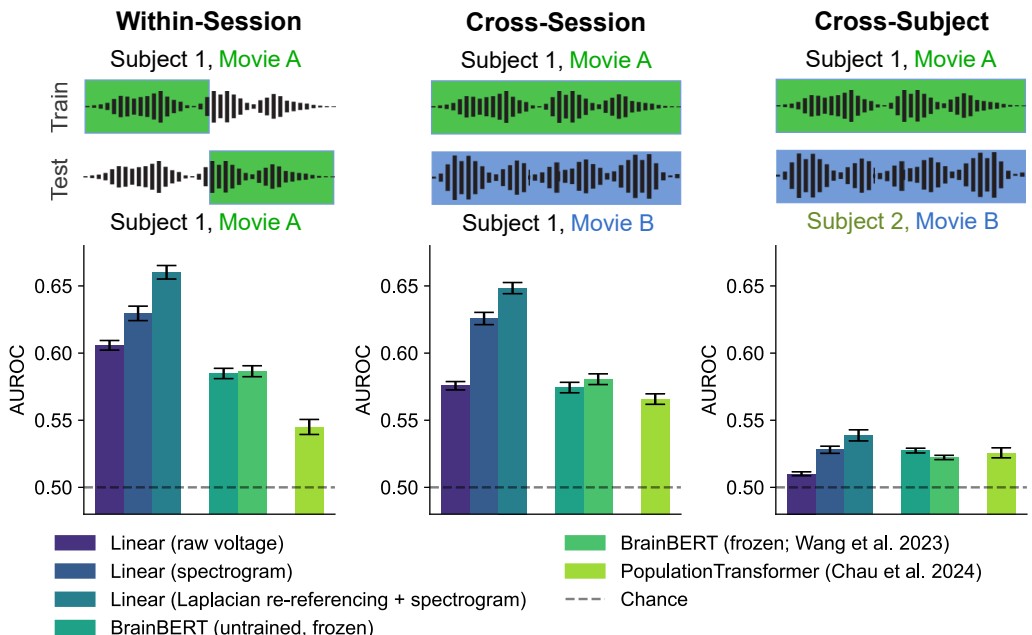

Figure 3: **Neuroprobe allows for evaluating decoding within and across recording sessions and subjects.** We perform analyses on three different types of splits (top row). In the *within-session* split, we train on data from one subject and one movie segment, and evaluate on the same subject, but another segment of the same movie. Performance is measured via cross-validation. In the *cross-session* split, we train and evaluate on different movies watched by the same subject. In the *cross-subject* split, we train on data from one subject and one movie and evaluate on data from an entirely different subject and movie. The cross-subject split is the most challenging for all evaluated baseline models (bottom row): (1) logistic regression either from raw voltage signal of all electrodes to the labels, or (2) from the spectrogram of the signal to the labels, including laplacian re-referencing (3), as well as (4) BrainBERT (Wang et al., 2023b) and (5) PopulationTransformer (Chau et al., 2024).

mann et al., 2012), but are limited in data quality and scale by today's standards. The NaturalScenes-Dataset (Allen et al., 2022) includes standardized splits, but uses fMRI data, a non-invasive modality that suffers from extremely low temporal resolution, and focuses mainly on visual processing. The clinical-grade Temple University Hospital EEG dataset (Obeid & Picone, 2016) can also be used as a benchmark, but it only contains scalp EEG data, and its labels are limited to seizure detection. Benchmarks for single-unit neural spiking data are proposed by Pei et al. (2021); Karpowicz et al. (2024); Willett et al. (2023); Lueckmann et al. (2025), but they only contain spiking information rather than broadband signals from iEEG that capture more aggregated neural activity (Parvizi & Kastner, 2018). To our knowledge, Neuroprobe is the first standardized benchmark for high fidelity intracranial EEG signals.

## 3 APPROACH

**The BrainTreebank dataset**    Neuroprobe uses the raw data from the BrainTreebank (Wang et al., 2024), a publicly available dataset released under a CC BY 4.0 license. The BrainTreebank is a large-scale dataset of intracranial electrophysiological recordings (stereoelectroencephalography; sEEG) collected while 10 human subjects (5 male, 5 female, ages 4–19; Supplementary Table 6) watched a total of 26 Hollywood movies (Supplementary Table 7). Electrode placements vary between patients, determined solely by the clinical needs of each neurosurgical patient, and are shown in Supplementary Figure 6. Spanning 43 hours of neural activity, the dataset aligns recorded brain signals with transcribed and manually corrected speech, word onsets, and universal dependency parses across the 223,068 words in 38,572 sentences.

**Decoding tasks**    We use the movie annotations and the alignment with the corresponding neural data to create a suite of 15 visual, audio, and language decoding tasks (Supplementary Section E). For every task, the input consists of a 1-second interval of neural data, starting at each word onset We found the results to be robust to jitter of the window start between -0.375-0.125s (Supplementary Figure 14). The annotation label is the target output. We formalize all of the tasks as binary classification by thresholding the labels according to their percentile in the full distribution of that type of annotation. For example, for the GPT2 Surprisal task, the positive label corresponds to surprisal annotations above the 75%th percentile of the distribution within a session, and the negative label to the values below the 25%th percentile. Moving beyond only binary tasks, we also offer the Multiclass version of Neuroprobe with the same 15 tasks (Supplementary Table 8). For non-scalar labels (such as part of speech of the word) we pick a main target class (i.e. *Verb* for the part of speech task), and formulate the task as one-versus-rest classification. Since we are studying realistic language processing with naturalistic stimuli, there are pre-existing relationships between the tasks in the movies. However, we found that these relations are actually very weak, (see Supplementary Figure 2); the average correlation between tasks is $r = 0.12 \pm 0.02$, averaged across all subjects and sessions. For more details, please see Supplementary Section E.

**Evaluation Splits**    The Neuroprobe benchmark supports three different types of splits (Figure 3):

1. **Within-Session**. In this split, training and test data both come from a single movie-viewing session. Decoding results are 2-fold cross-validated with 50-50 train/test splits. Importantly, the indices for the cross-validation splits are not drawn from the whole movie uniformly, but rather the train examples are taken from a single contiguous block and the validation examples are taken from a separate block. This is done to prevent models from overfitting to temporal auto-correlation (e.g. see Supplementary Figure I).

2. **Cross-Session**. The cross-session split even further ensures that no data contamination due to auto-correlation can occur, and tests the model's generalization to a novel recording session. The model is trained on data from one movie session and tested on another movie from the same subject. Unless otherwise noted, this is the split for most of the Neuroprobe results reported in this paper and will be the default on the leaderboard.

3. **Cross-Subject** This split evaluates the model's ability to generalize across subjects *and* stimuli. The training data consists of data from a single session (trial 4), viewed by subject 2, chosen because this is the longest trial in the dataset and since subject 2 contains the most electrodes in both hemispheres, allowing for maximum overlap with other subjects. Testing takes place using data from selected sessions for all other subjects (see Supplementary Section C). This split in particular presents a demanding test of generalization, especially since electrode placements vary widely between patients (see Supplementary Figure 6).

**Computational efficiency**    The full dataset of Neuroprobe *(Neuroprobe-Full)* allows flexibility for researchers to pick any of the 15 tasks and any of the 26 recording sessions in BrainTreebank. However, for the purposes of comparing models, running experiments over all sessions and electrodes is prohibitively expensive and unnecessary. So, when Neuroprobe is used as a *benchmark* (in text below, we refer to it simply as *Neuroprobe* when evaluating models), we subset the data to a smaller portion of subjects and recording sessions (6 subjects, 2 trials each, for a total of 12 sessions) for training and evaluation (Supplementary Section D).

Furthermore, to ensure computational efficiency, in the Neuroprobe benchmark, the total number of electrodes per subject is capped at 120, such that the input for each task is a standardized matrix which has predictable memory and computational requirements. The electrodes were selected in groups from complete probes to retain flexibiltiy for re-referencing techniques such as bipolar, common-average, or Laplacian re-referencing, which have been shown to improve the signal to noise ratio (Vidal et al., 2015; Li et al., 2018; Tsuchimoto et al., 2021). All selected electrodes have been localized in an average cortex atlas. To maximize the signal to noise ratio, the electrodes with the highest linear decoding performance were chosen first. The resulting standardized electrode subsets are available in the Neuroprobe codebase.

**Submissions and Leaderboard**    The primary evaluation metric is the Area Under the Receiver Operating Characteristic curve (AUROC). We will maintain a public leaderboard which will display model performance on this benchmark, both on the single-electrode and population level; see

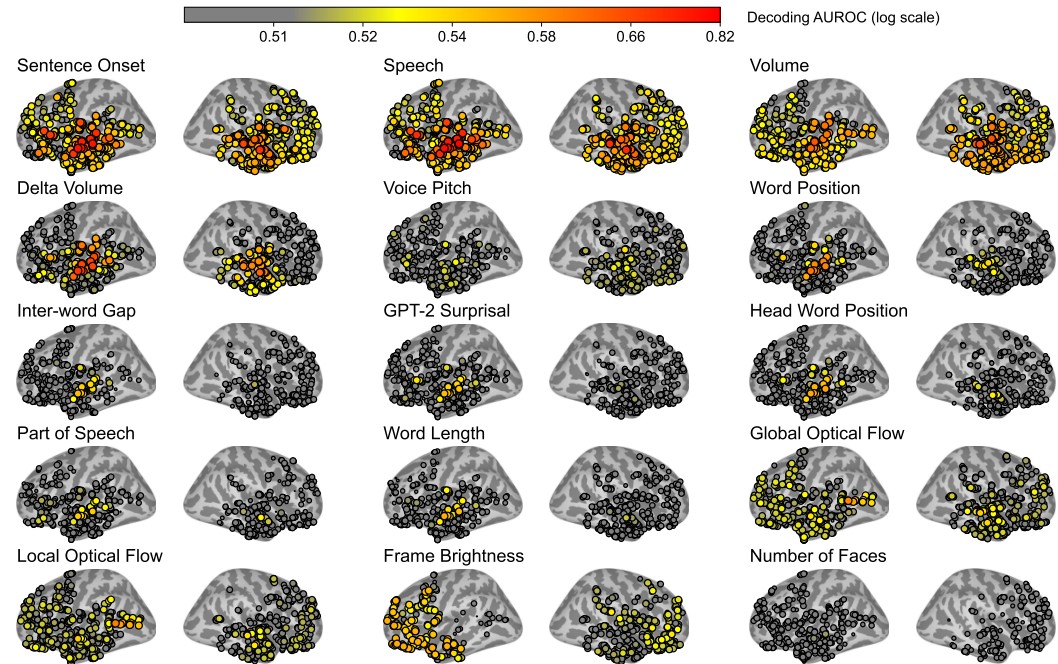

Figure 4: **Neuroprobe enables the visualization of how multimodal stimuli are processed throughout the brain.** This figure shows performance of linear decoders trained separately for every electrode's data on the *cross-session* split, averaged across all recording sessions of every subject. Color denotes AUROC on a logarithmic scale to show trends for tasks that have lower decodability. *Sentence Onset* is decodable throughout the brain, with a hotspot in the temporal lobe. Language features like *Part of Speech* and *GPT-2 Surprisal* are most decodable in the superior temporal lobe. Visual features such as *Optical Flow* are most decodable near the visual areas in the back of the brain, with some decodability in the frontal lobe.

Supplementary Figure 8. The evaluation rules and submission process is outlined in detail on the Neuroprobe website and in the code repository.

## 4 RESULTS

**Spatial analysis**   To investigate which brain areas are primarily involved in processing each Neuroprobe task, we examined the linear decodability of all Neuroprobe features (Figure 4). Using the single electrode analysis, we find that audio-linguistic tasks such as 'sentence onset', 'speech vs. non-speech', 'delta volume' are decodable at many sites in the brain, but the highest decoding performance is found in the superior temporal gyrus, especially close to Herschel's and Wernicke's area, with average AUROCs of $0.77$, $0.79$, and $0.69$, respectively in the gyrus of the temporal transverse. In contrast, visual features such as *Optical Flow* are most decodable near the visual areas in the occipital lobe, with some decodability in the frontal lobe. Here region results are given with respect to the Destrieux atlas; for more region-level analyses, see Supplementary Section N.

**Timing analysis**   To study the time course of linguistic information processing in the brain, we aligned neural data to word onsets and split it into narrow time-bins (width $= 250$ms), training a separate linear decoder on each bin for multiple tasks. Decodability is reported for the cross-session split. For each task, we restrict our attention to the top 100 electrodes with the highest decodability. Decoding performance as a function of time shows the course of processing after the word onset ($t = 0$, Figure 5). Interestingly, the beginning of a new sentence can be decoded with better-than-chance AUROC even before the word onset ($\mu = 0.55, \sigma_M = 0.002$ at $-250$ms), hinting at the predictive nature of processing. Moreover, we can find a time-ranking of features by looking at when decodability peaks for reach feature (Figure 5). For example, we note that the high-level

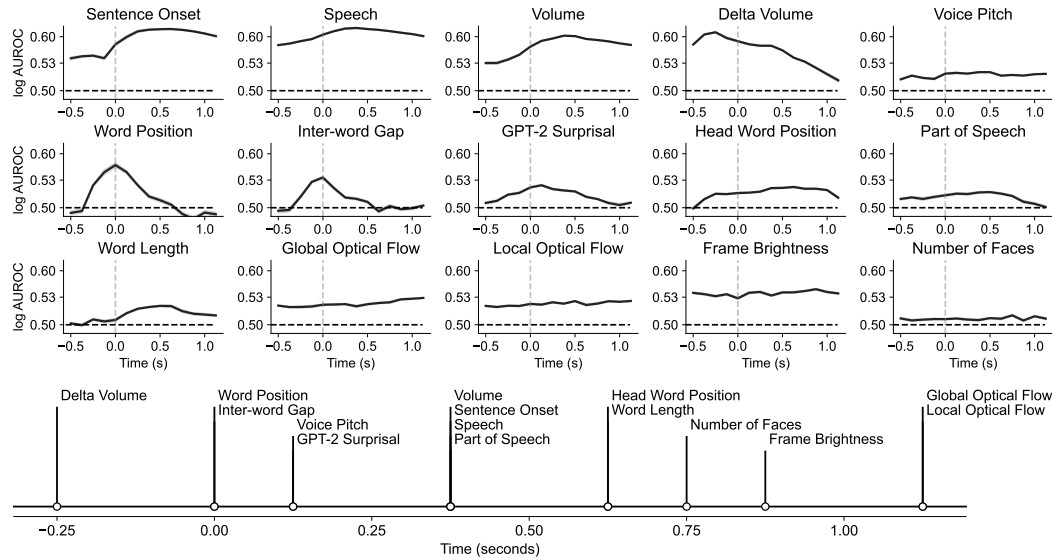

Figure 5: **Tracking multimodal sensory processing in the brain across time.** Here, we show the mean performance of the most decodable 100 electrodes per each task across time (top), where $t = 0$ corresponds to word onset. A linear model is fit on spectrograms of 250ms-long sliding windows of activity. Shaded regions denote s.e.m. across electrodes. We extract the peak of each decoding curve to obtain an approximate time ordering (bottom). Audio and linguistic features are most decodable close to word onset, whereas visual features like *Frame Brightness* and *Optical Flow* are most decodable around 1 second after word onset. Notably, *Head Word Position*, a semantic feature that pertains to the position of the dependency parse head, is decoded later than other language features. Note that we use a window which gives fairly coarse temporal localizations; in addition, these timings are dependent on the type of decoding analysis being performed, so the ordering may change once more advanced models are used.

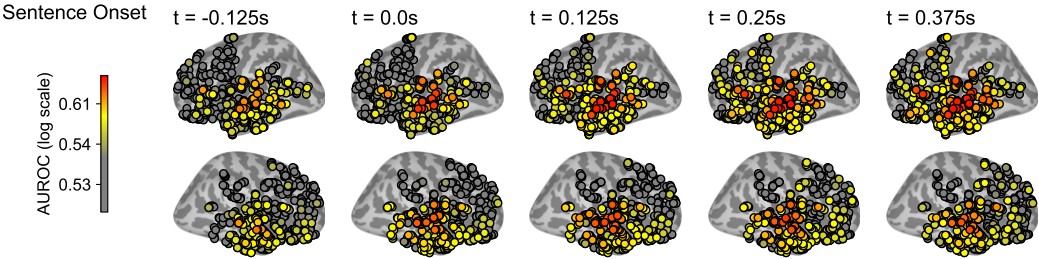

Figure 6: **Time evolution of speech onset decodability across brain areas.** The 'sentence onset' task is most decodable in the superior temporal gyrus at the first word's onset $(t = 0)$. Note that the decoding performance is above chance even before the speech onset, highlighting the predictive nature of sensory processing in the brain. As time progresses, speech becomes more decodable in the frontal areas of the brain as well, suggesting a flow of information from primary audio processing regions to the prefrontal cortex.

semantic feature 'word head position' is decodable only later (decodability peaks at $t = 0.625s$ vs. volume $t = 0.375s$ and pitch $t = 0.125s$).

**Spatio-Temporal analysis** We do a deep dive on the sentence-onset feature (Figure 6), investigating the time course of sentence onset decodability across brain areas. We found that right at the beginning of the sentence onset, it is most decodable in the temporal lobe (AUROC = 0.61 at $t = 0$ in the transverse temporal), but decodability spreads to the orbitofrontal cortex as time progresses

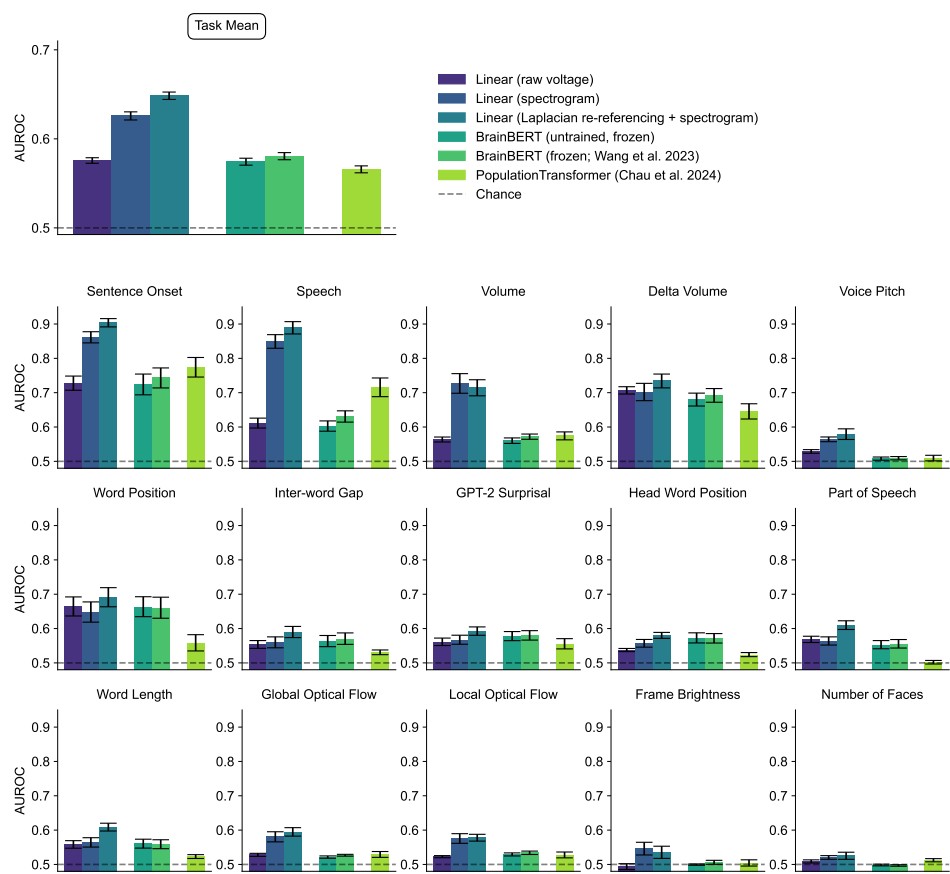

Figure 7: **Performance of baseline models on the 15 tasks of Neuroprobe (cross-session).** The performance of four models is displayed: (1) logistic regression either from raw voltage signal of all electrodes to the labels, or (2) from the spectrogram of the signal to the labels, including laplacian re-referencing (3), as well as (4) BrainBERT (Wang et al., 2023b) and (5) PopulationTransformer (Chau et al., 2024). For a rigorous and standardized evaluation, neural data was always cut to include one second following each word onset. Performance across different subjects and trials was averaged together. Error bars denote s.e.m. across all subjects and trials. These results can be seen in tabular form in Supplementary Section G.

(AUROC $= 0.51$ at $t = 0.0$ and AUROC $= 0.54$ at $t = 0.5$). We repeated this analysis for every task, generating maps of sensory processing: see Supplementary Figure 9 and Supplementary Figure 10.

**Comparison of basic decoding methods on Neuroprobe** To show the utility of the Neuroprobe as a benchmark, we establish baselines and evaluate frontier models. The models we benchmark span the range of simple classifiers to large, pretrained models.

The baselines include three linear regression models, which take as input either the raw voltage time-series inputs, spectrogram of the signal generated using the short-time Fourier transform (spectrogram), or the spectrogram of the Laplacian re-referenced inputs. We performed hyperparameter sweeps to determine the optimal spectrogram parameters, including number of data samples per STFT segment, percentage of overlap between consecutive segments, as well as the frequency range to keep; see Supplementary Figure 3 and Supplementary Figure 4). All inputs are given as a population, i.e., the data from all electrodes across all time samples is provided as input, concatenated.

We also decode using off-the-shelf representations from pretrained models, training a regression on single-channel BrainBERT (Wang et al., 2023b) inputs as well as the pretrained PopulationTrans-

former (Chau et al., 2024), a pretrained transformer for encoding arbitrary sets of electrode activities. More details on the models available in Supplementary Section F.

Perhaps surprisingly, we found that linear decoding on spectrogram inputs with Laplacian-rereferencing is a very strong baseline (see Supplementary Figure 3), achieving the best overall performance on the within-session ($0.660 \pm 0.005$), cross session ($0.648 \pm 0.004$), and cross-subject split ($0.539 \pm 0.004$). This shows the importance of optimizing the spectrogram parameters. In comparison, linear decoding on raw voltage achieves ($0.510 \pm 0.001$) on the cross-subject split, while BrainBERT improves slightly over this ($0.522 \pm 0.002$). In general, the aggregated BrainBERT representations result in the next best decoding: $0.586 \pm 0.004$ on within-session and $0.581 \pm 0.004$ on cross-session. Meanwhile, PopT achieves $0.545 \pm 0.006$ and $0.566 \pm 0.004$ on both splits respectively.

Finally, for the cross-session split, a breakdown by task can be seen in Figure 7. The Population-Transformer, despite being pretrained, underperforms on many tasks, but achieves good performance on the Sentence Onset and Speech vs. Non-speech tasks.

## 5 CONCLUSION

Neuroprobe can be used in several ways by different communities. Machine learning researchers can treat it as any other benchmark and build decoding models that directly optimize for classification performance. Meanwhile, practitioners at the intersection of ML and neuroscience can build foundation models or virtual brains based on principled neuroscience priors and use Neuroprobe to measure improvements in the learned representations. Finally, neuroscientists can use Neuroprobe on its own or in tandem with models from the first two communities to uncover relationships between different aspects of multi-modal sensory processing in the brain. We hope that Neuroprobe will both drive improvements in decoding and in our ability to draw neuroscience conclusions from large scale data. Furthermore, as we have seen in other fields, it can also lead to a virtuous cycle in which neuroscientists are encouraged to develop and release more open neural datasets to the effort.

Perhaps surprisingly, we found that the linear baseline with spectrogram inputs provides a very strong baseline, outperforming foundation models on many tasks, highlighting the need for a standardized benchmark to drive progress. Even using this simple baseline, Neuroprobe yields insights into both the spatial and temporal organization of tasks in the brain. As decoding models improve, the clarity of such findings will improve and their variance will decrease.

It is our hope that Neuroprobe will drive significant advances in iEEG foundation models by providing a standardized, multi-task evaluation that encourages development of more performant architectures. These improved foundation models could translate into meaningful clinical benefits, including more precise brain-computer interfaces that offer finer motor control for patients with paralysis, more accurate seizure prediction algorithms that provide earlier intervention opportunities, and deeper insights into language processing that could inform rehabilitation strategies for stroke and brain injury patients, potentially accelerating the development of next-generation neural prosthetics and therapeutic interventions that could dramatically improve quality of life for patients with neurological conditions.

**Limitations**   While the BrainTreebank dataset endows Neuroprobe with unprecedented combination of scale and resolution, it is collected from a clinical population undergoing invasive monitoring, and results may not be overgeneralized. At the moment, the dataset only contains 10 subjects. This low number of subjects is due to the fact that iEEG data is difficult to get, as it requires invasive surgery to implant the electrodes. However, this is a difficulty faced by the field at large; for example, the widely used Natural Scenes Dataset Allen et al. (2022) has 8 subjects.

**Future work**   Our framework is general enough to accommodate future annotations, allowing for investigations of low-level language processing, such as syllable-count, or high-level semantic processing such as thematic roles or language model embeddings. We seek, in near-term future work, to add to the library of tasks and datasets in Neuroprobe. As we continue to build our benchmark, researchers will be able to study the question of how various tasks interact with each other.

**Broader impacts** Neuroprobe provides a standardized benchmark for evaluating models of human brain activity, with potential applications in neuroscience, machine learning, and clinical technologies such as brain-computer interfaces. By releasing our data, code, and leaderboard, we aim to democratize access to high-quality neural benchmarks and enable measurable progress in the field of iEEG foundation models.

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

## A  LLM USAGE

LLMs were not used in the ideation or for the bulk of the writing of this work. They were primarily used to polish the writing on a per-phrase basis.

## B  ETHICS STATEMENTS

The authors have read and adhered to the ICLR code of ethics. Our work uses existing, publicly available and anonymized human data.

**Reproducibility** We publicly release the code required to produce our decoding results. The leaderboard will be hosted online. Submissions will be unverified except for formatting, however, we will require an accompanying publication and link to a code repository for every submission in order to receive a "reproducible" designation on the leaderboard.

## C  SPLITS

Neuroprobe includes 3 different types of splits.

**Within-Session** In this split, models are trained and tested within the same subject and the same movie session. To avoid temporal data leakage, we are using 2-fold cross-validation using non-overlapping segments of the movie. We found that 2-fold cross-validation yields virtually identical results to 5-fold cross-validation, while having a 60% lower computational load ($r = 0.982, p < 0.001$, Supplementary Figure 1).

**Cross-Session** This is a slightly more difficult split. It ensures completely that no data-contamination due to auto-correlation has occurred. The model is being trained on data from one movie session and tested on another movie from the same subject.

**Cross-Subject** This is the most difficult split. It tests the model's ability to generalize between subjects *and* stimuli. Specifically, the model is trained exclusively on Subject 2, Trial 4 (Guardians of the Galaxy 2), and evaluated independently on all other subjects and each of their movie sessions. This is especially challenging considering the variability in electrode placements per subject. Our current approach for adapting the linear baselines includes initially pre-processing neural data to represent activity in each cortical region (using averaging per subject/trial pair), as defined from the 34 regions by the Desikan-Killany atlas (Desikan et al., 2006). For every pair of subjects, we only consider those atlas regions that are present in both subjects. Then, we evaluated different linear baselines on the preprocessed data.

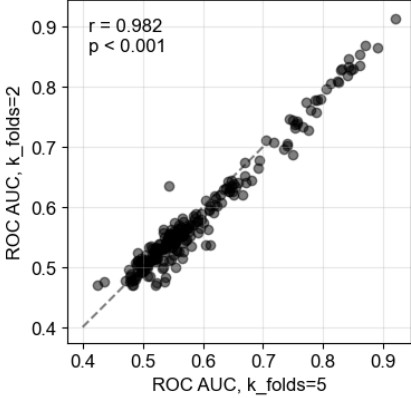

Supplementary Figure 1: **Extremely high correlation between 2-fold and 5-fold cross-validation results on Neuroprobe, within-session split.**

## D  NEUROPROBE-LITE

The following subject-trial pairs are included in Neuroprobe Lite:

- Subject 1: Trials 1, 2
- Subject 2: Trials 0, 4
- Subject 3: Trials 0, 1
- Subject 4: Trials 0, 1
- Subject 7: Trials 0, 1
- Subject 10: Trials 0, 1

For every task, the number of datapoints was trimmed at 3500 datapoints (i.e. if a specific movie has more than 3500 annotations for any task, only 3500 are taken total for the Lite benchmark). When selecting the subject/trial pairs for Neuroprobe Lite, we selected the trials that contained the most tasks which hit the 3500 datapoints limit. The sampling is done in a stratified way, sampling

equal number of samples per class, and guaranteeing in this way that the classes are balanced. So, for example, in a 2-class task, "taking the first 3500 annotations" means specifically "take the first 1750 samples of class 1 and the first 1750 samples of class 2", which guarantees the class balance by design.

Furthermore, to ensure computational efficiency, the total number of electrodes per subject is capped at 120, such that the input for each task is a standardized matrix which has predictable memory and computational requirements. The electrodes were selected in groups from complete probes to retain flexibiblity for re-referencing techniques such as bipolar, common-average, or Laplacian re-referencing, which have been shown to improve the signal to noise ratio (Vidal et al., 2015; Li et al., 2018; Tsuchimoto et al., 2021). All selected electrodes have been localized in an average cortex atlas. To maximize the signal to noise ratio, the electrodes with the highest linear decoding performance were chosen first.

# E DECODING TASKS

| # | Feature | Description | Benchmark Task |
|---|---------|-------------|----------------|
| 1 | frame_brightness *(visual)* | The mean brightness computed as the average HSV value over all pixels | Binary classification: low (percentiles 0%-25%) vs high (75%-100%) |
| 2 | global_flow *(visual)* | A camera motion proxy. The maximal average dense optical flow vector magnitude | Same as above |
| 3 | local_flow *(visual)* | A large displacement proxy. The maximal optical flow vector magnitude | Same as above |
| 4 | face_num *(visual)* | The maximum number of faces per frame during the word | 2-way classification: 0, or $\geq 1$ |
| 5 | volume *(auditory)* | Average root mean squared watts of the audio | Binary classification: low (0%-25%) vs high (75%-100%) |
| 6 | pitch *(auditory)* | Average pitch of the audio | Same as above |
| 7 | delta_volume *(auditory)* | The difference in average RMS of the 500ms windows pre- and post-word onset | Same as above |
| 8 | speech *(language)* | Whether any speech is present in the given time interval | Binary classification |
| 9 | onset *(language)* | Whether a new sentence starts in the interval, or there is no speech at all | Binary classification |
| 10 | gpt2_surprisal *(language)* | Negative-log transformed GPT-2 word probability (given preceding 20s of language context) | Binary classification: low (0%-25%) vs high (75%-100%) |
| 11 | word_length *(language)* | Word length (ms) | Same as above |
| 12 | word_gap *(language)* | Difference between previous word offset and current word onset (ms) | Same as above |
| 13 | word_index *(language)* | The word index in its context sentence | 2-way classification: 0 (the first word in the sentence), or other (1) |
| 14 | word_head_pos *(language)* | The relative position (left/right) of the word's dependency tree head | Binary classification |
| 15 | word_part_speech *(language)* | The word Universal Part-of-Speech (UPOS) tag | 2-way classification: verb (0), or other (1) |

Supplementary Table 1: **Extracted visual, auditory, and language features used to create the evaluations for Neuroprobe.** For all classification tasks, the classes were rebalanced. The difference between local and global flow is that global is the averaged optical flow, with the average being taken over all optical flow vectors on the screen, whereas local is the largest individual optical flow vector on the screen. The table is adapted from Chau et al. (2024).

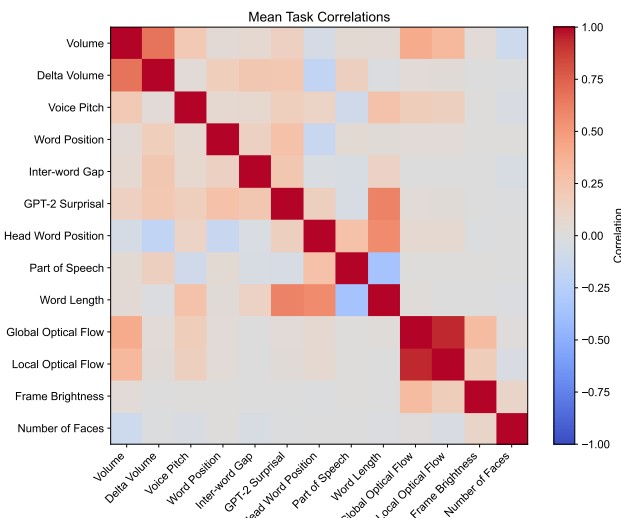

Supplementary Figure 2: **Correlations between tasks** Averaged across movies, the off-diagonal correclation between tasks is $0.121 \pm 0.019$. Note that the tasks Speech and Sentence Onset are not represented here, because they do not share the same underlying data samples (specifically, when the label is 0 for those tasks, it means that there is no speech in the movie, and many of the other tasks are undefined).

# F MODELS BENCHMARKED

**Linear (raw voltage)**   For this evaluation, raw voltage traces from the BrainTreebank data sampled at 2048 Hz were fed as input to the linear regression. We found almost identical results when removing line noise or passing the data raw to the linear regression, so the raw data was used in the paper. When removing line noise, it was removed at $60 \pm 5$ Hz and the 4 harmonics,

**Linear (spectrogram)**   For this baseline evaluation, the features are the spectrogram of the raw signal with the following parameters (given that the sampling rate is 2048Hz):

- nperseg=512
- noverlap=75%
- window=hann
- Frequency range: 0-150Hz.

After this step, the data turns into an array of arrays where first dimension is the time bin and the second dimension is the spectrogram result across frequencies; for the downstream regression, all of these features are concatenated together.

We performed sweeps to determine the optimal hyperparameters for the spectrogram (number of data samples per STFT segment, percentage of overlap between consecutive segments, as well as max and min frequency to retain; see Supplementary Figure 3 and Supplementary Figure 4).

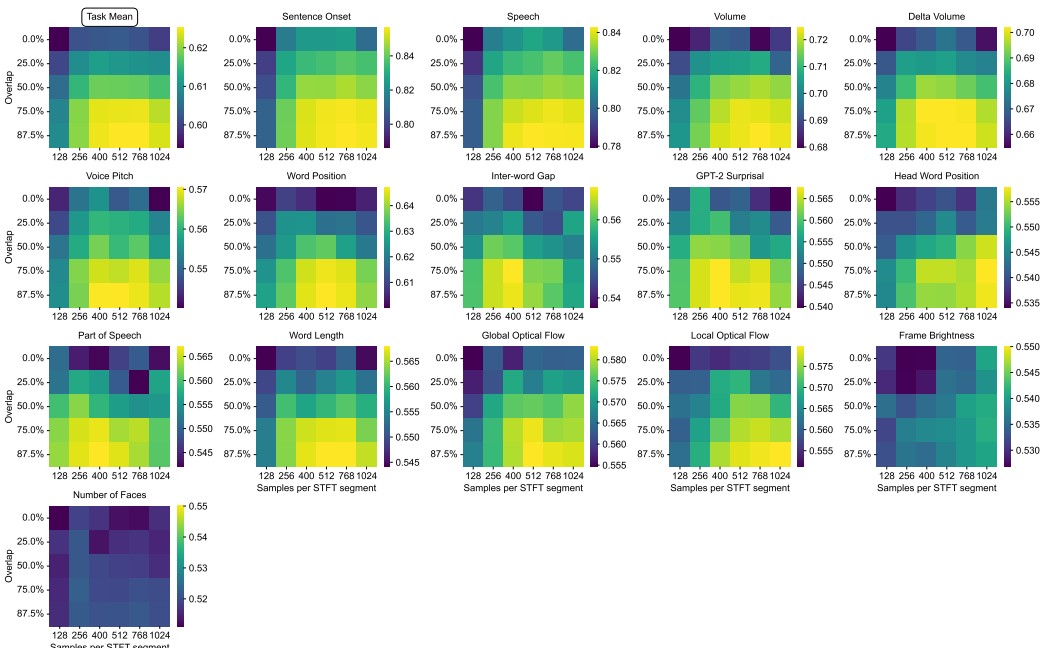

Supplementary Figure 3: **A sweep of the spectrogram hyperparameters: data samples per STFT segment and the overlap between consecutive segments.**

**BrainBERT**   For this evaluation, the BrainTreebank data was Laplacian rereferenced (as described in the original BrainBERT paper by Wang et al. (2023b)), with line noise removed, and then passed into the BrainBERT model as provided by Wang et al. (2023b). The output features were concatenated and used as input to the linear regression. For the electrodes which could not be Laplacian rereferenced, non-rereferenced data was inputted into BrainBERT. The BrainBERT model was frozen and only the final linear regression layer was fine tuned, in order to compare the quality of features generated by the foundation model.

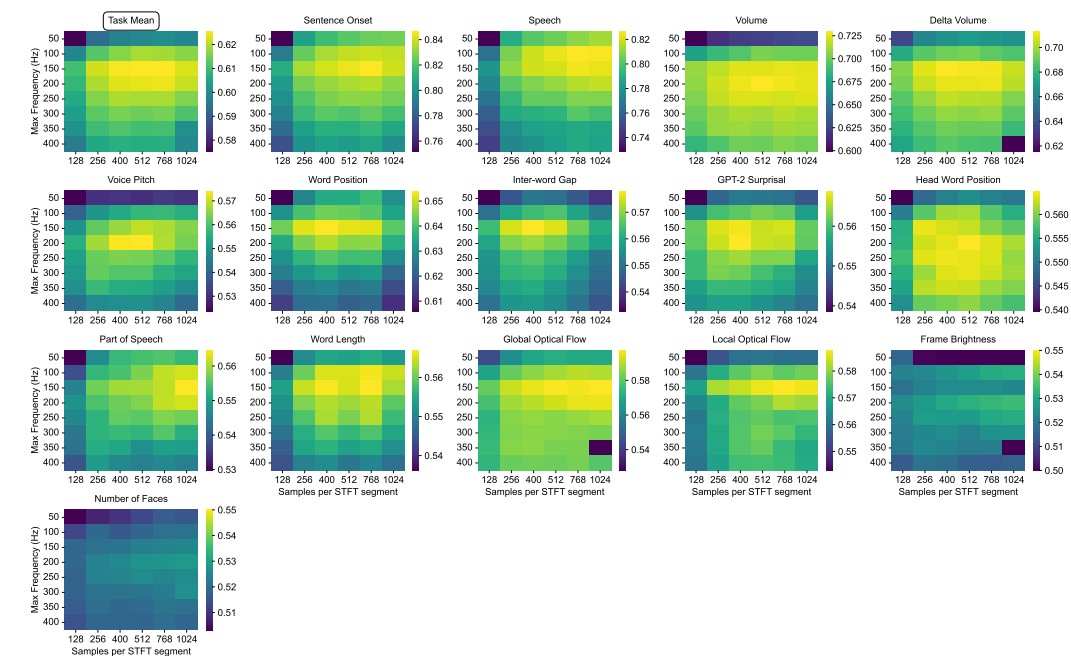

Supplementary Figure 4: **A sweep of the spectrogram hyperparameters: data samples per STFT segment and the maximum frequency that is included as part of the feature vector.** The analysis is done using the 75% overlap between consecutive STFT segments.

For all linear regression, we used the sklearn package, class LinearRegression, with the tolerance parameter set as 0.001. In all cases, the features were first normalized using the sklearn Standard-Scaler. We found that it helps with convergence and often produces higher regression values for the baselines.

**Population Transformer** Population Transformer (PopT) is a SSL pretrained model for encoding arbitrary ensembles of iEEG electrode data for general downstream decoding (Chau et al., 2024). The model consists of a transformer backbone that learns functional and spatial relationships between input channels whose temporal activity is encoded. We use the publicly available weights which were pretrained on data from 10 iEEG subjects, using 5s BrainBERT temporal embeddings from individual channels. For PopT, we followed the implementation and used the weights from (Chau et al., 2024). The fine-tuning protocol is taken to be directly the same as in the authors' original paper (including linear rate, number of epochs, a factor of 10 between learning rates of the linear output layer vs the transformer blocks, etc), but reduce the number of steps to $steps = 1000$. We finetune PopT in two conditions: either by only finetuning the final linear output layer while keeping the rest of the model weights frozen (the "frozen" condition), or finetuning through the whole model (the default PopT condition).

When running linear regressions on the cross-subject splits, in order to arrive at a subject-agnostic input, we represent neural activity using a single average vector per region for each of the 34 regions by the Desikan-Killiany atlas (Desikan et al., 2006). We use this same scheme when running cross-subject decoding with BrainBERT. No accommodation for the cross-subject split was necessary for the PopulationTransformer, which is designed to handle subject-transfer. For the PopulationTransformer, we use only those electrodes in the Neuroprobe subset that can be Laplacian-rereferenced and are in the set of 'clean' electrodes (see Chau et al. (2024)) for evaluation.

# G BENCHMARK RESULTS

## G.1 WITHIN-SESSION SPLITS

| Model | Overall | Sentence Onset | Speech | Volume |
|---|---|---|---|---|
| Linear (voltage) | $0.606 \pm 0.004$ | $0.795 \pm 0.021$ | $0.656 \pm 0.022$ | $0.595 \pm 0.015$ |
| Linear (spectrogram) | $0.630 \pm 0.005$ | $0.851 \pm 0.025$ | $0.825 \pm 0.028$ | $0.726 \pm 0.038$ |
| Linear (Laplacian+spectrogram) | $\mathbf{0.660 \pm 0.005}$ | $\mathbf{0.891 \pm 0.018}$ | $\mathbf{0.883 \pm 0.018}$ | $\mathbf{0.717 \pm 0.032}$ |
| BrainBERT (untrained, frozen) | $0.585 \pm 0.004$ | $0.750 \pm 0.028$ | $0.603 \pm 0.020$ | $0.570 \pm 0.008$ |
| BrainBERT (frozen) | $0.586 \pm 0.004$ | $0.757 \pm 0.027$ | $0.611 \pm 0.022$ | $0.583 \pm 0.010$ |
| PopulationTransformer | $0.545 \pm 0.006$ | $0.689 \pm 0.050$ | $0.677 \pm 0.044$ | $0.576 \pm 0.018$ |

| Model | Delta Volume | Voice Pitch | Word Position | Inter-word Gap |
|---|---|---|---|---|
| Linear (voltage) | $0.753 \pm 0.019$ | $0.536 \pm 0.005$ | $\mathbf{0.742 \pm 0.017}$ | $0.595 \pm 0.015$ |
| Linear (spectrogram) | $0.718 \pm 0.025$ | $0.570 \pm 0.011$ | $0.657 \pm 0.029$ | $0.579 \pm 0.019$ |
| Linear (Laplacian+spectrogram) | $\mathbf{0.762 \pm 0.026}$ | $\mathbf{0.578 \pm 0.016}$ | $0.740 \pm 0.028$ | $\mathbf{0.612 \pm 0.014}$ |
| BrainBERT (untrained, frozen) | $0.697 \pm 0.020$ | $0.524 \pm 0.005$ | $0.684 \pm 0.027$ | $0.583 \pm 0.017$ |
| BrainBERT (frozen) | $0.706 \pm 0.021$ | $0.524 \pm 0.007$ | $0.685 \pm 0.027$ | $0.584 \pm 0.017$ |
| PopulationTransformer | $0.628 \pm 0.025$ | $0.509 \pm 0.008$ | $0.519 \pm 0.023$ | $0.509 \pm 0.009$ |

| Model | GPT-2 Surprisal | Head Word Position | Part of Speech | Word Length |
|---|---|---|---|---|
| Linear (voltage) | $0.584 \pm 0.009$ | $0.570 \pm 0.008$ | $0.576 \pm 0.012$ | $0.599 \pm 0.013$ |
| Linear (spectrogram) | $0.570 \pm 0.017$ | $0.565 \pm 0.012$ | $0.559 \pm 0.011$ | $0.569 \pm 0.017$ |
| Linear (Laplacian+spectrogram) | $\mathbf{0.613 \pm 0.017}$ | $\mathbf{0.602 \pm 0.012}$ | $\mathbf{0.605 \pm 0.012}$ | $\mathbf{0.618 \pm 0.015}$ |
| BrainBERT (untrained, frozen) | $0.581 \pm 0.013$ | $0.587 \pm 0.012$ | $0.553 \pm 0.010$ | $0.571 \pm 0.012$ |
| BrainBERT (frozen) | $0.580 \pm 0.015$ | $0.585 \pm 0.013$ | $0.556 \pm 0.012$ | $0.571 \pm 0.013$ |
| PopulationTransformer | $0.523 \pm 0.014$ | $0.519 \pm 0.008$ | $0.513 \pm 0.004$ | $0.505 \pm 0.005$ |

| Model | Global Optical Flow | Local Optical Flow | Frame Brightness | Number of Faces |
|---|---|---|---|---|
| Linear (voltage) | $0.535 \pm 0.009$ | $0.544 \pm 0.005$ | $0.507 \pm 0.013$ | $0.499 \pm 0.007$ |
| Linear (spectrogram) | $0.604 \pm 0.017$ | $0.593 \pm 0.020$ | $\mathbf{0.533 \pm 0.015}$ | $0.525 \pm 0.008$ |
| Linear (Laplacian+spectrogram) | $\mathbf{0.625 \pm 0.013}$ | $\mathbf{0.607 \pm 0.017}$ | $0.521 \pm 0.025$ | $\mathbf{0.530 \pm 0.014}$ |
| BrainBERT (untrained, frozen) | $0.528 \pm 0.005$ | $0.528 \pm 0.003$ | $0.504 \pm 0.005$ | $0.505 \pm 0.005$ |
| BrainBERT (frozen) | $0.521 \pm 0.006$ | $0.525 \pm 0.003$ | $0.508 \pm 0.012$ | $0.503 \pm 0.007$ |
| PopulationTransformer | $0.509 \pm 0.008$ | $0.508 \pm 0.014$ | $0.499 \pm 0.019$ | $0.492 \pm 0.010$ |

Supplementary Table 2: Performance comparison across tasks (mean $\pm$ SEM) on the within-session split. Best performing model for each task is shown in bold.

## G.2 CROSS-SESSION SPLITS

| Model | Overall | Sentence Onset | Speech | Volume |
|---|---|---|---|---|
| Linear (voltage) | $0.576 \pm 0.003$ | $0.728 \pm 0.021$ | $0.611 \pm 0.014$ | $0.564 \pm 0.007$ |
| Linear (spectrogram) | $0.626 \pm 0.005$ | $0.861 \pm 0.016$ | $0.849 \pm 0.020$ | $0.727 \pm 0.029$ |
| Linear (Laplacian+spectrogram) | $\mathbf{0.648 \pm 0.004}$ | $\mathbf{0.904 \pm 0.012}$ | $\mathbf{0.889 \pm 0.018}$ | $\mathbf{0.714 \pm 0.023}$ |
| BrainBERT (untrained, frozen) | $0.574 \pm 0.004$ | $0.724 \pm 0.030$ | $0.603 \pm 0.015$ | $0.560 \pm 0.008$ |
| BrainBERT (frozen) | $0.581 \pm 0.004$ | $0.743 \pm 0.029$ | $0.631 \pm 0.016$ | $0.572 \pm 0.008$ |
| PopulationTransformer | $0.566 \pm 0.004$ | $0.774 \pm 0.028$ | $0.716 \pm 0.027$ | $0.574 \pm 0.012$ |

| Model | Delta Volume | Voice Pitch | Word Position | Inter-word Gap |
|---|---|---|---|---|
| Linear (voltage) | $0.707 \pm 0.011$ | $0.529 \pm 0.005$ | $0.664 \pm 0.028$ | $0.554 \pm 0.011$ |
| Linear (spectrogram) | $0.702 \pm 0.025$ | $0.564 \pm 0.007$ | $0.648 \pm 0.029$ | $0.560 \pm 0.016$ |
| Linear (Laplacian+spectrogram) | $\mathbf{0.734 \pm 0.020}$ | $\mathbf{0.579 \pm 0.016}$ | $\mathbf{0.691 \pm 0.028}$ | $\mathbf{0.590 \pm 0.016}$ |
| BrainBERT (untrained, frozen) | $0.680 \pm 0.019$ | $0.508 \pm 0.005$ | $0.664 \pm 0.029$ | $0.564 \pm 0.016$ |
| BrainBERT (frozen) | $0.692 \pm 0.020$ | $0.509 \pm 0.005$ | $0.661 \pm 0.031$ | $0.571 \pm 0.017$ |
| PopulationTransformer | $0.646 \pm 0.022$ | $0.510 \pm 0.008$ | $0.559 \pm 0.024$ | $0.531 \pm 0.007$ |

| Model | GPT-2 Surprisal | Head Word Position | Part of Speech | Word Length |
|---|---|---|---|---|
| Linear (voltage) | $0.561 \pm 0.011$ | $0.537 \pm 0.005$ | $0.569 \pm 0.009$ | $0.558 \pm 0.011$ |
| Linear (spectrogram) | $0.567 \pm 0.013$ | $0.557 \pm 0.011$ | $0.564 \pm 0.012$ | $0.564 \pm 0.014$ |
| Linear (Laplacian+spectrogram) | $\mathbf{0.593 \pm 0.012}$ | $\mathbf{0.580 \pm 0.009}$ | $\mathbf{0.610 \pm 0.013}$ | $\mathbf{0.609 \pm 0.011}$ |
| BrainBERT (untrained, frozen) | $0.578 \pm 0.013$ | $0.573 \pm 0.015$ | $0.553 \pm 0.012$ | $0.561 \pm 0.013$ |
| BrainBERT (frozen) | $0.580 \pm 0.014$ | $0.572 \pm 0.014$ | $0.556 \pm 0.012$ | $0.559 \pm 0.013$ |
| PopulationTransformer | $0.556 \pm 0.015$ | $0.524 \pm 0.006$ | $0.502 \pm 0.005$ | $0.523 \pm 0.006$ |

| Model | Global Optical Flow | Local Optical Flow | Frame Brightness | Number of Faces |
|---|---|---|---|---|
| Linear (voltage) | $0.528 \pm 0.004$ | $0.523 \pm 0.003$ | $0.494 \pm 0.009$ | $0.509 \pm 0.005$ |
| Linear (spectrogram) | $0.580 \pm 0.015$ | $0.576 \pm 0.014$ | $\mathbf{0.546 \pm 0.018}$ | $0.520 \pm 0.005$ |
| Linear (Laplacian+spectrogram) | $\mathbf{0.595 \pm 0.012}$ | $\mathbf{0.578 \pm 0.010}$ | $0.535 \pm 0.018$ | $\mathbf{0.525 \pm 0.010}$ |
| BrainBERT (untrained, frozen) | $0.521 \pm 0.003$ | $0.529 \pm 0.004$ | $0.500 \pm 0.002$ | $0.498 \pm 0.002$ |
| BrainBERT (frozen) | $0.527 \pm 0.003$ | $0.534 \pm 0.005$ | $0.506 \pm 0.006$ | $0.497 \pm 0.002$ |
| PopulationTransformer | $0.529 \pm 0.008$ | $0.528 \pm 0.009$ | $0.504 \pm 0.009$ | $0.512 \pm 0.005$ |

Supplementary Table 3: Performance comparison across tasks (mean $\pm$ SEM) on the cross-session split. Best performing model for each task is shown in bold.

## G.3 CROSS-SUBJECT SPLITS

| Model | Overall | Sentence Onset | Speech | Volume |
|---|---|---|---|---|
| Linear (voltage) | $0.510 \pm 0.001$ | $0.539 \pm 0.013$ | $0.508 \pm 0.006$ | $0.513 \pm 0.004$ |
| Linear (spectrogram) | $0.528 \pm 0.003$ | $0.621 \pm 0.024$ | $0.585 \pm 0.018$ | $0.530 \pm 0.008$ |
| Linear (Laplacian+spectrogram) | $\mathbf{0.539 \pm 0.004}$ | $\mathbf{0.673 \pm 0.037}$ | $\mathbf{0.642 \pm 0.038}$ | $\mathbf{0.527 \pm 0.010}$ |
| BrainBERT (untrained, frozen) | $0.527 \pm 0.002$ | $0.585 \pm 0.014$ | $0.537 \pm 0.007$ | $0.524 \pm 0.003$ |
| BrainBERT (frozen) | $0.522 \pm 0.002$ | $0.582 \pm 0.013$ | $0.537 \pm 0.005$ | $0.521 \pm 0.003$ |
| PopulationTransformer | $0.526 \pm 0.004$ | $0.638 \pm 0.031$ | $0.594 \pm 0.035$ | $0.526 \pm 0.012$ |

| Model | Delta Volume | Voice Pitch | Word Position | Inter-word Gap |
|---|---|---|---|---|
| Linear (voltage) | $0.533 \pm 0.010$ | $0.503 \pm 0.002$ | $0.539 \pm 0.009$ | $0.511 \pm 0.003$ |
| Linear (spectrogram) | $0.555 \pm 0.011$ | $0.505 \pm 0.005$ | $0.552 \pm 0.013$ | $0.508 \pm 0.006$ |
| Linear (Laplacian+spectrogram) | $0.568 \pm 0.013$ | $0.505 \pm 0.002$ | $0.571 \pm 0.022$ | $0.515 \pm 0.008$ |
| BrainBERT (untrained, frozen) | $\mathbf{0.590 \pm 0.010}$ | $0.505 \pm 0.003$ | $\mathbf{0.574 \pm 0.015}$ | $0.513 \pm 0.003$ |
| BrainBERT (frozen) | $0.574 \pm 0.010$ | $0.507 \pm 0.002$ | $0.549 \pm 0.012$ | $0.510 \pm 0.003$ |
| PopulationTransformer | $0.573 \pm 0.016$ | $\mathbf{0.509 \pm 0.005}$ | $0.503 \pm 0.007$ | $\mathbf{0.519 \pm 0.005}$ |

| Model | GPT-2 Surprisal | Head Word Position | Part of Speech | Word Length |
|---|---|---|---|---|
| Linear (voltage) | $0.510 \pm 0.005$ | $0.504 \pm 0.003$ | $0.495 \pm 0.003$ | $0.502 \pm 0.003$ |
| Linear (spectrogram) | $0.510 \pm 0.004$ | $0.511 \pm 0.004$ | $0.509 \pm 0.003$ | $0.509 \pm 0.003$ |
| Linear (Laplacian+spectrogram) | $0.508 \pm 0.003$ | $0.521 \pm 0.008$ | $0.508 \pm 0.006$ | $0.508 \pm 0.005$ |
| BrainBERT (untrained, frozen) | $\mathbf{0.522 \pm 0.005}$ | $\mathbf{0.530 \pm 0.005}$ | $\mathbf{0.517 \pm 0.003}$ | $\mathbf{0.509 \pm 0.004}$ |
| BrainBERT (frozen) | $0.511 \pm 0.004$ | $0.524 \pm 0.004$ | $0.509 \pm 0.004$ | $0.504 \pm 0.004$ |
| PopulationTransformer | $0.522 \pm 0.007$ | $0.509 \pm 0.007$ | $0.498 \pm 0.005$ | $0.498 \pm 0.004$ |

| Model | Global Optical Flow | Local Optical Flow | Frame Brightness | Number of Faces |
|---|---|---|---|---|
| Linear (voltage) | $0.500 \pm 0.004$ | $0.500 \pm 0.002$ | $0.493 \pm 0.002$ | $0.501 \pm 0.003$ |
| Linear (spectrogram) | $0.508 \pm 0.007$ | $0.506 \pm 0.009$ | $\mathbf{0.514 \pm 0.008}$ | $0.496 \pm 0.003$ |
| Linear (Laplacian+spectrogram) | $\mathbf{0.515 \pm 0.005}$ | $\mathbf{0.513 \pm 0.005}$ | $0.499 \pm 0.004$ | $\mathbf{0.508 \pm 0.004}$ |
| BrainBERT (untrained, frozen) | $0.503 \pm 0.003$ | $0.501 \pm 0.006$ | $0.502 \pm 0.002$ | $0.500 \pm 0.003$ |
| BrainBERT (frozen) | $0.501 \pm 0.002$ | $0.498 \pm 0.004$ | $0.506 \pm 0.004$ | $0.501 \pm 0.004$ |
| PopulationTransformer | $0.503 \pm 0.007$ | $0.500 \pm 0.010$ | $0.502 \pm 0.009$ | $0.494 \pm 0.004$ |

Supplementary Table 4: Performance comparison across tasks (mean $\pm$ SEM) on the cross-subject split. Best performing model for each task is shown in bold.

# H  SUBJECT AND MOVIE INFORMATION

| Subj. | Age (yrs.) | # Electrodes | Movie | Recording time (hrs) | Neuroprobe |
|---|---|---|---|---|---|
| 1 | 19 | 154 | Fantastic Mr. Fox | 1.35 | |
| | | | The Martian | 2.43 | x |
| | | | Thor: Ragnarok | 1.77 | x |
| 2 | 12 | 162 | Venom | 1.54 | x |
| | | | Spider-Man: Homecoming | 2.05 | |
| | | | Guardians of the Galaxy | 1.90 | |
| | | | Guardians of the Galaxy 2 | 2.13 | x |
| | | | Avengers: Infinity War | 2.30 | |
| | | | Black Panther | 1.42 | |
| | | | Aquaman | 2.19 | |
| 3 | 18 | 134 | Cars 2 | 1.64 | x |
| | | | Lord of the Rings 1 | 2.25 | x |
| | | | Lord of the Rings 2 (extended edition) | 3.58 | |
| 4 | 12 | 188 | Shrek 3 | 1.38 | x |
| | | | Megamind | 1.44 | x |
| | | | Incredibles | 0.85 | |
| 5 | 6 | 156 | Fantastic Mr. Fox | 1.35 | |
| 6 | 9 | 164 | Megamind | 0.68 | |
| | | | Toy Story | 1.29 | |
| | | | Coraline | 0.84 | |
| 7 | 11 | 246 | Cars 2 | 1.64 | x |
| | | | Megamind | 1.44 | x |
| 8 | 4.5 | 162 | Sesame Street Episode | 0.94 | |
| 9 | 16 | 106 | Ant Man | 1.80 | |
| 10 | 12 | 216 | Cars 2 | 1.33 | x |
| | | | Spider-Man: Far from Home | 1.93 | x |

Supplementary Table 5: **Subject statistics** Subjects in the BrainTreebank dataset, and the trials used in the benchmark tasks. Table adapted from Wang et al. (2023b). The second column shows the total number of electrodes. The average amount of recording data per subject is 4.3 (hrs).

| Subj. | Age | Sex | Movies | Time (h) | # Sent. | # Words | # Lemmas | # Elec. | # Probes |
|---|---|---|---|---|---|---|---|---|---|
| 1 | 19 | M | 7, 18, 19 | 5.6 | 4372 | 27424 | 4489 | 154 | 13 |
| 2 | 12 | M | 2, 3, 4, 8, 9, 17, 21 | 13.5 | 9870 | 57731 | 9164 | 162 | 47 |
| 3 | 18 | F | 5, 11, 12 | 7.5 | 5281 | 31596 | 4547 | 134 | 12 |
| 4 | 12 | F | 10, 13, 15 | 3.7 | 4056 | 23876 | 4017 | 188 | 15 |
| 5 | 6 | M | 7 | 1.35 | 1282 | 7908 | 1481 | 156 | 12 |
| 6 | 9 | F | 6, 13, 20 | 2.8 | 3789 | 20089 | 3349 | 164 | 12 |
| 7 | 11 | F | 5, 13 | 3.08 | 3523 | 19068 | 2828 | 246 | 18 |
| 8 | 4 | M | 14 | 0.94 | 860 | 3994 | 537 | 162 | 13 |
| 9 | 16 | F | 1 | 1.80 | 1558 | 9235 | 1480 | 106 | 12 |
| 10 | 12 | M | 5, 16 | 3.08 | 3981 | 22147 | 3004 | 216 | 17 |

Supplementary Table 6: **All subjects language, electrodes and personal statistics.** Columns from left to right are the subject's ID and information (age and gender), the IDs of the movies they watched (corresponding to Supplementary Table 7), the cumulative movie time (hours), number of sentences, number of words (tokens) and number of unique lemmas (canonical word forms), as well as the number of probes the subject had and their corresponding number of electrodes. Table adapted from Wang et al. (2024).

| # Movie | Year | Length | Sent. | Words | Unique words | Nouns | Unique nouns | Verbs | Unique verbs |
|---|---|---|---|---|---|---|---|---|---|
| 1 Antman | 2015 | 7027 | 1558 | 9869 | 1944 | 1358 | 705 | 1545 | 580 |
| 2 Aquaman | 2018 | 8601 | 1054 | 7233 | 1544 | 1069 | 520 | 1104 | 508 |
| 3 Avengers: Infinity War | 2018 | 8961 | 1523 | 8529 | 1750 | 1083 | 607 | 1317 | 495 |
| 4 Black Panther | 2018 | 8073 | 1254 | 7580 | 1606 | 1093 | 553 | 1209 | 508 |
| 5 Cars 2 | 2011 | 6377 | 2051 | 11407 | 2037 | 1572 | 724 | 1664 | 577 |
| 6 Coraline | 2009 | 6036 | 997 | 5433 | 1232 | 784 | 409 | 805 | 348 |
| 7 Fantastic Mr. Fox | 2009 | 5205 | 1282 | 8461 | 1864 | 1229 | 681 | 1227 | 484 |
| 8 Guardians of the Galaxy 1 | 2014 | 7251 | 1174 | 8295 | 1779 | 1096 | 603 | 1250 | 529 |
| 9 Guardians of the Galaxy 2 | 2017 | 8146 | 1290 | 9405 | 1824 | 1224 | 626 | 1370 | 532 |
| 10 Incredibles | 2003 | 6926 | 1521 | 9430 | 1954 | 1226 | 652 | 1557 | 591 |
| 11 Lord of the Rings 1 | 2001 | 13699 | 1514 | 10566 | 1998 | 1473 | 679 | 1487 | 598 |
| 12 Lord of the Rings 2 | 2002 | 14131 | 1716 | 11041 | 2065 | 1588 | 743 | 1619 | 646 |
| 13 Megamind | 2010 | 5735 | 1472 | 8891 | 1726 | 1172 | 602 | 1347 | 496 |
| 14 Sesame Street Ep. 3990 | 2016 | 3440 | 860 | 4220 | 787 | 717 | 231 | 706 | 217 |
| 15 Shrek the Third | 2007 | 5568 | 1063 | 7226 | 1590 | 977 | 568 | 1071 | 422 |
| 16 Spiderman: Far From Home | 2019 | 7764 | 1930 | 12189 | 1969 | 1459 | 668 | 1785 | 560 |
| 17 Spiderman: Homecoming | 2017 | 8008 | 2196 | 12295 | 2066 | 1583 | 777 | 1808 | 572 |
| 18 The Martian | 2015 | 9081 | 1570 | 11374 | 2192 | 1757 | 812 | 1677 | 622 |
| 19 Thor: Ragnarok | 2017 | 7831 | 1583 | 9683 | 1789 | 1195 | 599 | 1419 | 548 |
| 20 Toy Story 1 | 1995 | 4863 | 1320 | 7216 | 1510 | 1019 | 548 | 1027 | 395 |
| 21 Venom | 2018 | 6727 | 1379 | 7937 | 1513 | 897 | 507 | 1217 | 433 |

Supplementary Table 7: **Language statistics for all movies.** Columns from left to right are the movie's ID, name, year of production, length (seconds), number of sentences, number of words (tokens), number of unique words (types), number of nouns, number of unique nouns, number of verbs and number of unique verbs. Table adapted from Wang et al. (2024).

# I COMPOSITION OF MOVIES BY VOLUME

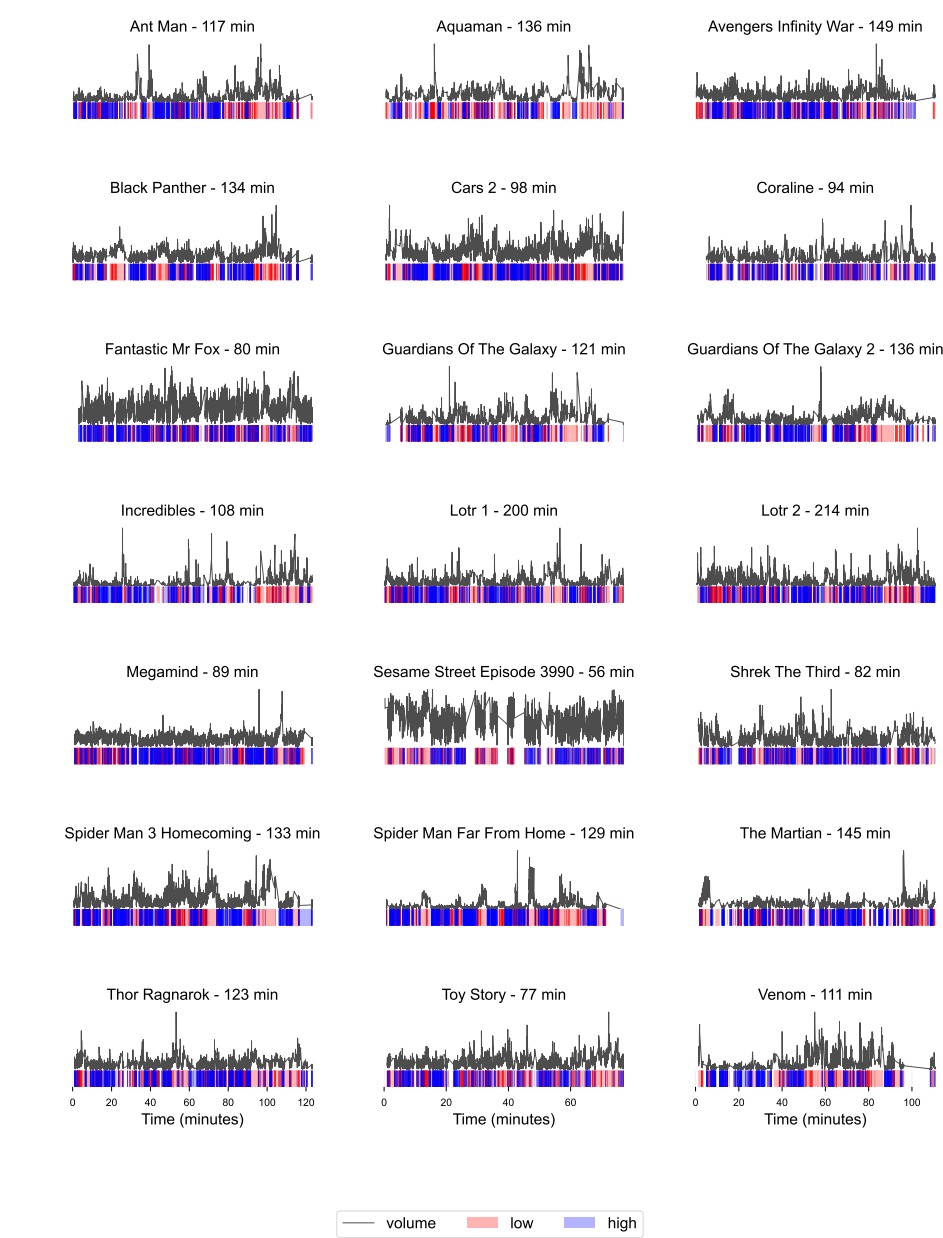

Supplementary Figure 5: **Volume comparison across movies.** The black line shows the normalized audio volume over time for 18 feature-length films and one TV episode shown to subjects. Below each volume trace, colored bars indicate periods of relatively low (red) and high (blue) volume, defined as the bottom $25\%$ and top $25\%$ of volume values respectively.

## J    ELECTRODE LOCATIONS

Subject 1   (N=3 sessions; N=130 electrodes)

Subject 2   (N=7 sessions; N=135 electrodes)

Subject 3   (N=3 sessions; N=124 electrodes)

Subject 4   (N=3 sessions; N=185 electrodes)

Subject 5   (N=1 sessions; N=140 electrodes)

Subject 6   (N=3 sessions; N=161 electrodes)

Subject 7   (N=2 sessions; N=240 electrodes)

Subject 8   (N=1 sessions; N=153 electrodes)

Subject 9   (N=1 sessions; N=99 electrodes)

Subject 10   (N=2 sessions; N=207 electrodes)

Supplementary Figure 6: **Electrode locations across subjects.** Brain reconstructions showing electrode placement and speech-selective responses for all 10 subjects. Each dot represents an electrode. Only non-corrupted electrodes are included in this figure.

# K FACE DISTRIBUTION

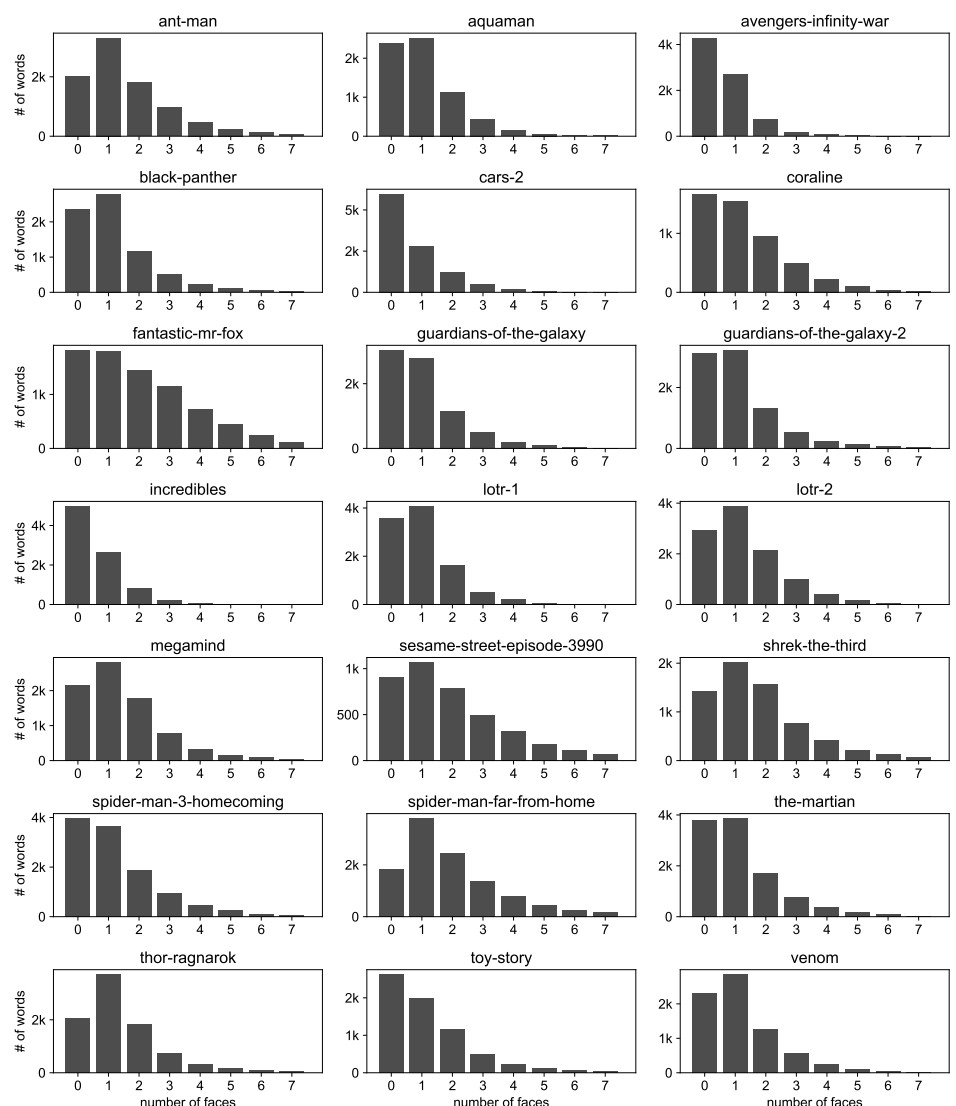

Supplementary Figure 7: **Distribution of faces detected per frame across different movies.** Histograms show the number of words (y-axis) that occur during frames containing different numbers of faces (x-axis) for 18 feature-length films and one TV episode (Sesame Street) used in BrainTreebank.

# L COMPUTE REQUIREMENTS

Every Linear regression was run on a CPU-only instance, with 2 virtual CPU cores and 64GB RAM for the population level results and 2 CPU cores with 6GB RAM for the single electrode decoding results. For BrainBERT, the necessary resources also included a GPU with at least 9GB of memory along with 128GB of RAM and 2 CPU cores. For the PopulationTransformer, the fine-tuning was done on 2 GPUs (NVIDIA GeForce GTX TITAN X) with at least 12GB of GPU RAM.

# M LEADERBOARD

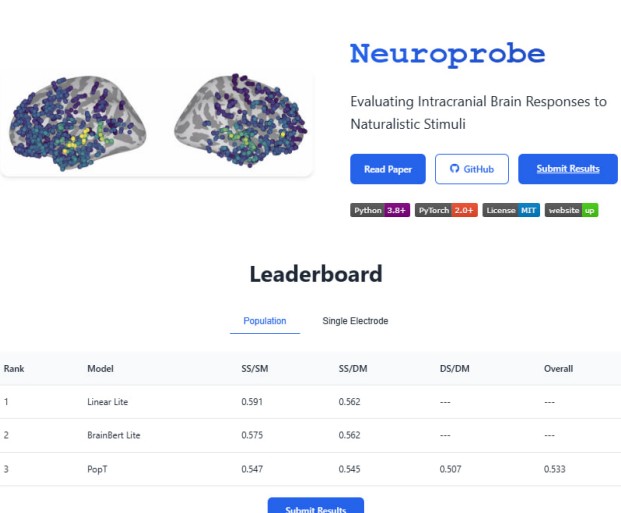

Supplementary Figure 8: **The leaderboard for the task of classifying sentence onset.** The public webpage link will be made available upon publication. Submissions will be submitted to our github repository. Once accepted, the performance numbers will be displayed on the public leaderboard. Submissions will consist of either the single-electrode-level or population-level performances. Submitters can choose to submit either one or both. Leaderboard placement will be determined by results on the cross-session split, but the other splits will be displayed as well.

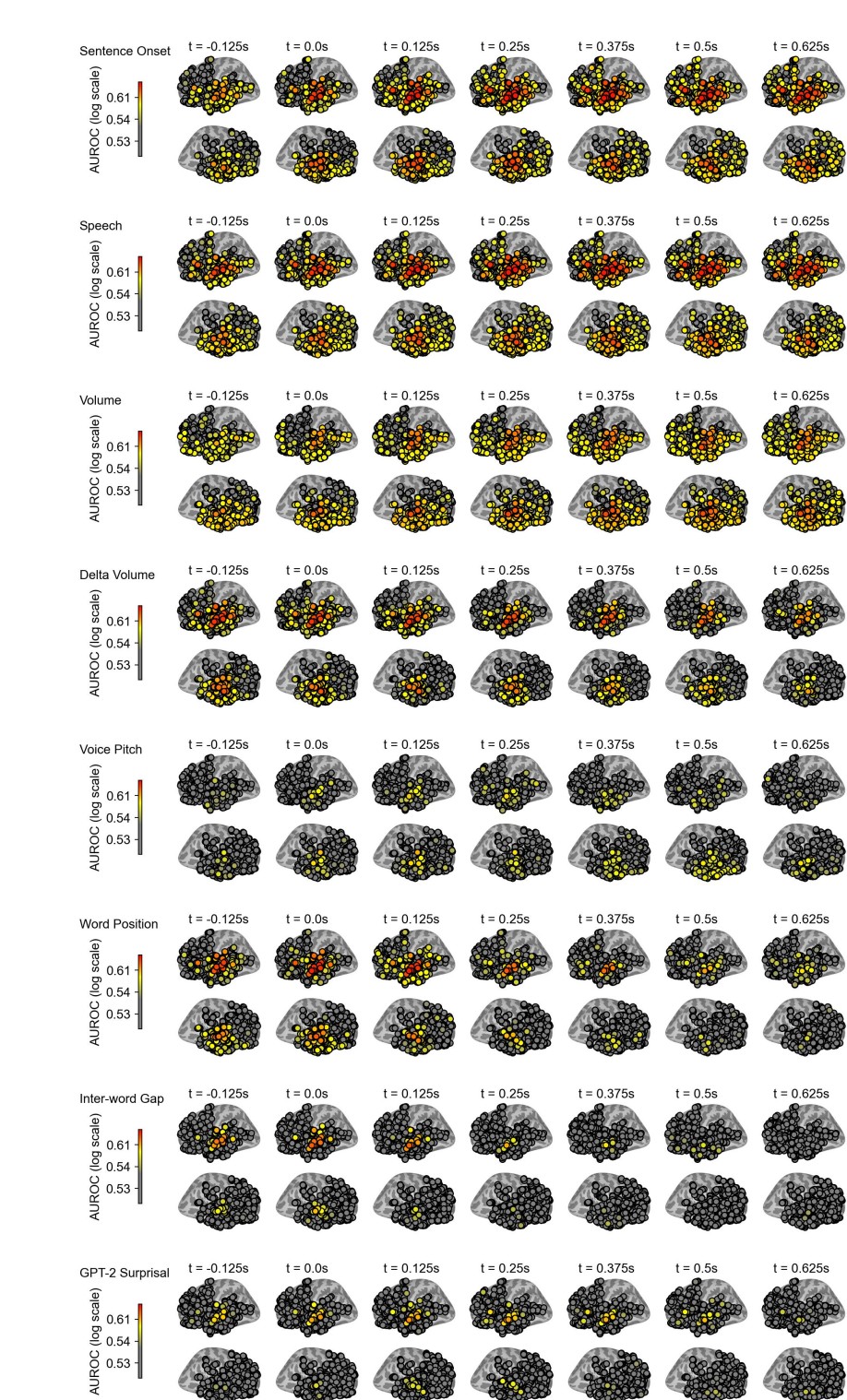

Supplementary Figure 9:   **Spatio-temporal course of decodability**  This is the same information as Figure 6, but for all tasks. Each row shows the spatio-temporal course of decodability for a given task. Each column shows one time slice.

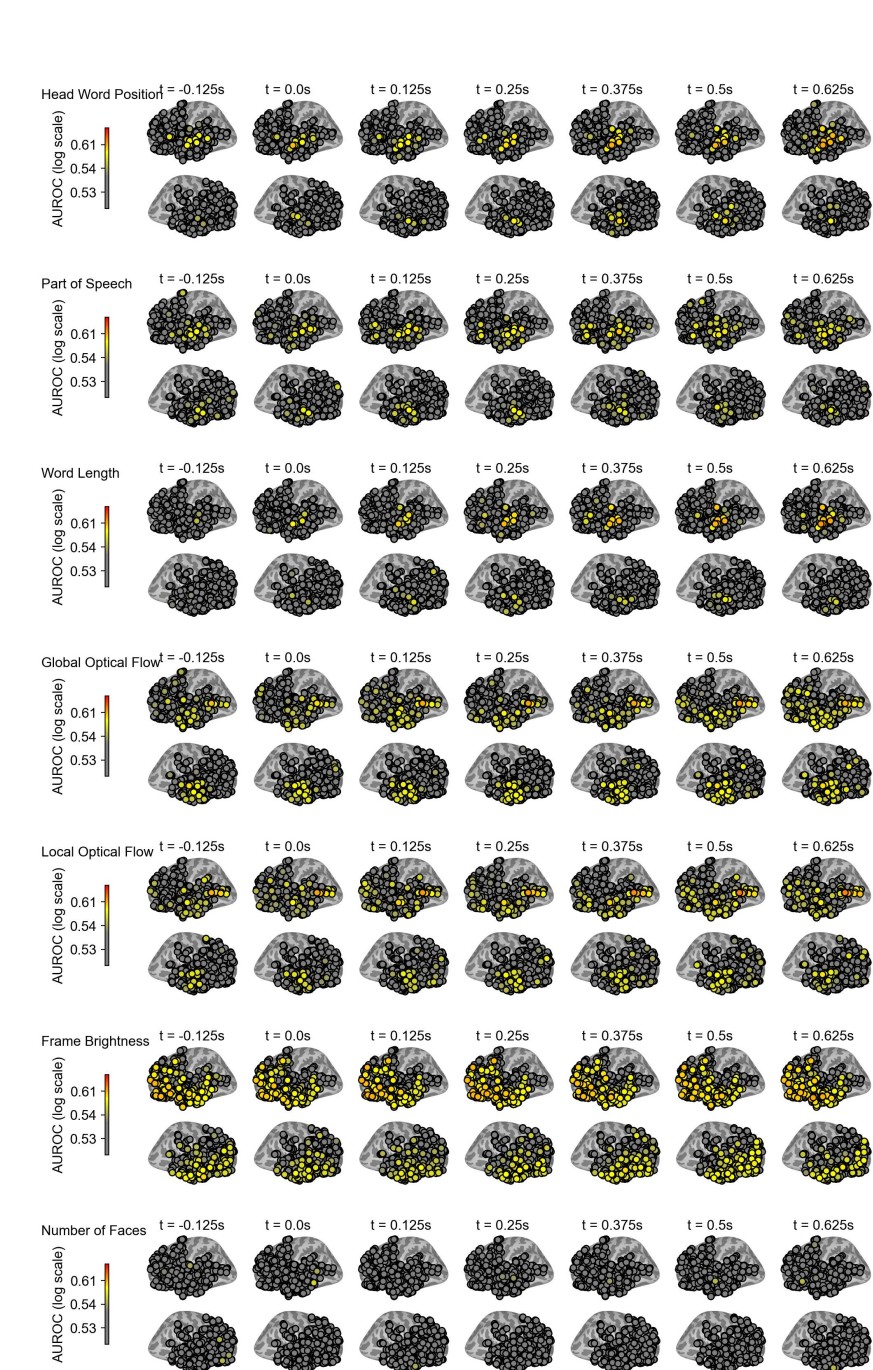

Supplementary Figure 10: Supplementary Figure 9 continued.

# N REGION ANALYSIS

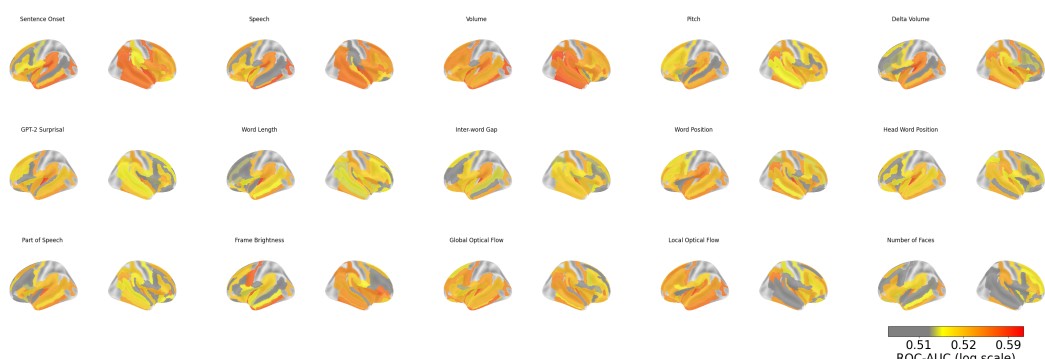

Supplementary Figure 11: The same information as in Figure 4 is displayed, but aggregated according to the Destrieux atlas.

## O    MULTICLASS NEUROPROBE TASKS

| # | Feature | Description | Benchmark Task |
|---|---------|-------------|----------------|
| 1 | frame_brightness *(visual)* | The mean brightness computed as the average HSV value over all pixels | 3-way classification: low (percentiles 0%-25%) vs medium (37.5%-62.5%) vs high (75%-100%) |
| 2 | global_flow *(visual)* | A camera motion proxy. The maximal average dense optical flow vector magnitude | Same as above |
| 3 | local_flow *(visual)* | A large displacement proxy. The maximal optical flow vector magnitude | Same as above |
| 4 | face_num *(visual)* | The maximum number of faces per frame during the word | 3-way classification: 0, or 1, or $> 1$ |
| 5 | volume *(auditory)* | Average root mean squared watts of the audio | 3-way classification: low (percentiles 0%-25%) vs medium (37.5%-62.5%) vs high (75%-100%) |
| 6 | pitch *(auditory)* | Average pitch of the audio | Same as above |
| 7 | delta_volume *(auditory)* | The difference in average RMS of the 500ms windows pre- and post-word onset | Same as above |
| 8 | speech *(language)* | Whether any speech is present in the given time interval | Binary classification |
| 9 | onset *(language)* | Whether a new sentence starts in the interval, or there is no speech at all | Binary classification |
| 10 | gpt2_surprisal *(language)* | Negative-log transformed GPT-2 word probability (given preceding 20s of language context) | 3-way classification: low (percentiles 0%-25%) vs medium (37.5%-62.5%) vs high (75%-100%) |
| 11 | word_length *(language)* | Word length (ms) | Same as above |
| 12 | word_gap *(language)* | Difference between previous word offset and current word onset (ms) | Same as above |
| 13 | word_index *(language)* | The word index in its context sentence | 3-way classification: 0 (the first word in the sentence), 1 (second word), or other |
| 14 | word_head_pos *(language)* | The relative position (left/right) of the word's dependency tree head | Binary classification |
| 15 | word_part_speech *(language)* | The word Universal Part-of-Speech (UPOS) tag | 6-way classification: noun (0), or verb (1), or pronoun (2), determiner (3), adjective (4), adverb (5). |

Supplementary Table 8: **Neuroprobe-Multiclass: the multiclass classification versions of the extracted visual, auditory, and language tasks in Neuroprobe.** For all classification tasks, the classes were rebalanced.

# P    ADDITIONAL TABLES AND FIGURES

| Avg. test ROC-AUC | IOU | Subject |
|---|---|---|
| 0.578849 | 0.041667 | sub_1 |
| 0.517130 | 0.071429 | sub_10 |
| 0.540566 | 0.371429 | sub_3 |
| 0.501422 | 0.476190 | sub_7 |

Supplementary Table 9: **Performance and region overlap with held-out subject in cross-subject split**. Region overlap is quantified as intersection-over-union (IOU) between Desikan-Killiany regions that are sampled between the train subject (subject 4), and the test subjects. The performance is the averaged AUC-ROC across all tasks and trials.

| Model | # Parameters |
|---|---|
| PopulationTransformer | 20M |
| Linear Baseline | 77K |
| BrainBERT Regression | 3M |

Supplementary Table 10: **Parameter counts for baseline models**. Note that BrainBERT is the parameters in the regression over BrainBERT embeddings, not the actual BrainBERT model.

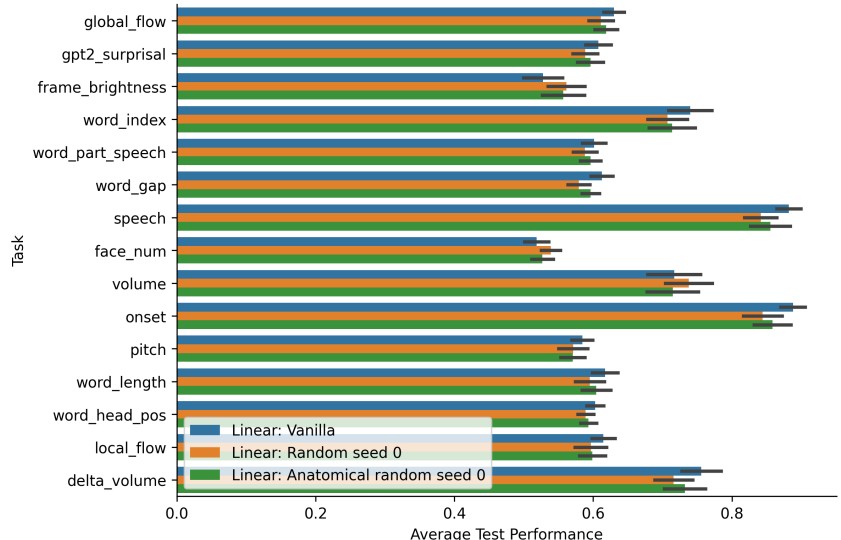

Supplementary Figure 12: **Task performance of linear baseline across different electrode selections** We show the linear decoder performance on subsets of electrodes which were selected according to decodability (blue), uniformly at random without replacement (orange), and according to anatomy – frontal and temporal electrodes.

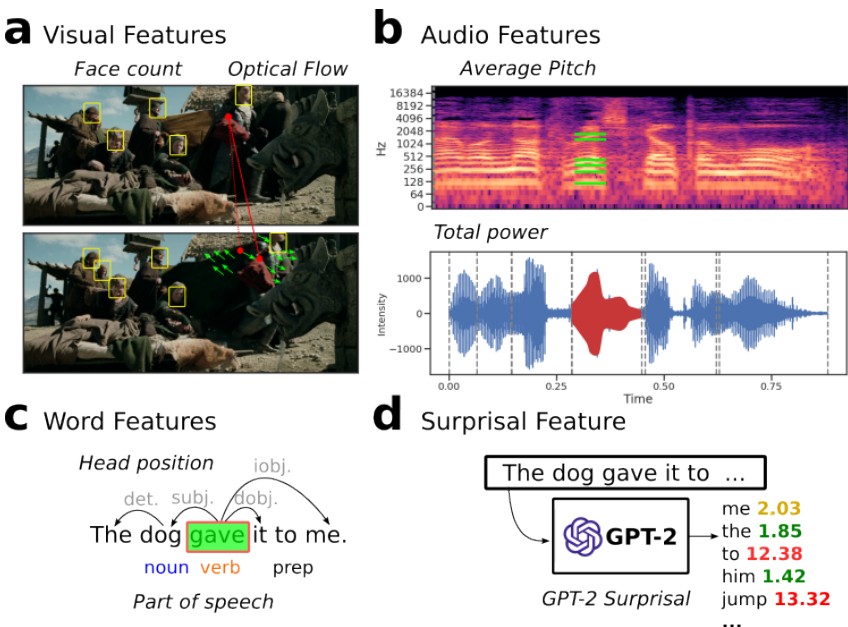

Supplementary Figure 13: **Schematic overview** of selected visual (**a**), audio (**b**), and language (**c-d**) elements of Neuroprobe. Visual features (**a**) include the number of faces (yellow boxes), and the magnitude and angle of optical flow (green arrows). Audio features (**b**) include the average pitch (top) and volume (bottom) during each word. Word features (**c**) include part-of-speech and the position of each word's dependency head. A surprisal feature, (**d**), computed using GPT-2 (**?**), a large language model, is the negative log probability of the word given the preceding context. Figure and caption adapted from Wang et al. (2024) and included here for reference.

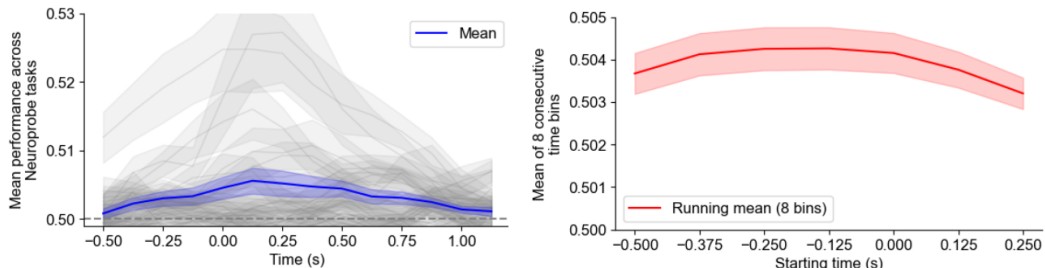

Supplementary Figure 14: **Robustness of Neuroprobe performance across the window jitter.** We show that across tasks, the decodability peaks around 0.0-0.25 seconds after the word onset (Left; gray lines correspond to different tasks, and the blue line is the mean across tasks). Across all tasks, the one-second interval can start anywhere from -0.375seconds to 0.0 seconds relative to the word onset, and the decoding performance is roughly identical. These results are obtained by running Neuroprobe decoding using data from one electrode at a time and averaging the results.

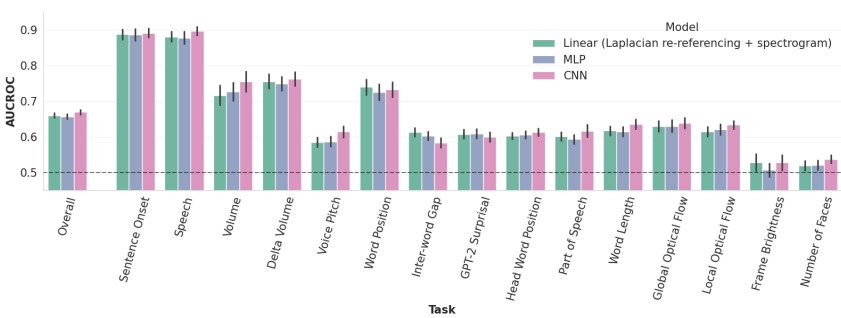

Supplementary Figure 15: **Additional baselines**. We train a CNN (pink) and MLP (purple) on the Neuroprobe tasks for the Within-Session split with the same spectrogram inputs as Linear (green). The CNN (pink) achieves the best performance by a slight margin. The MLP has 2 hidden layers each with 128 dimensions. The CNN has 3 conv layers, each with kernel size=3, and channel out=32, 64, and 128. It is followed by a fully connected hidden layer with 128 dimensions before outputting to a single value.

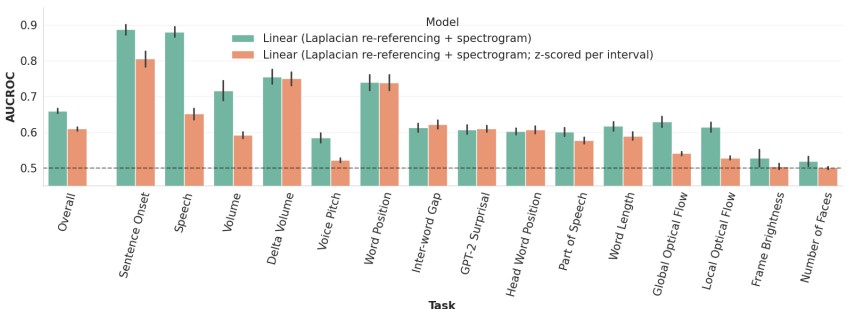

Supplementary Figure 16: **The importance of pre-processing** Here, we show the sensitivity of decoding to input normalization. In our Linear baseline, we z-scored the data across the whole train dataset (green). In contrast, z-scoring inputs within a 1-second interval results in lower performance (green). This is notable because this latter type of normalization was the scheme used by BrainBERT (Wang et al., 2023a) and the PopulationTransformer (Chau et al., 2024), which partially explains the difference in performance between the linear baseline and the pretrained models.

| Model | Overall | Sentence Onset | Speech | Volume |
|---|---|---|---|---|
| Linear (voltage) | $0.554 \pm 0.002$ | $0.728 \pm 0.021$ | $0.611 \pm 0.014$ | $0.530 \pm 0.003$ |
| Linear (spectrogram) | $0.593 \pm 0.003$ | $0.861 \pm 0.016$ | $0.849 \pm 0.020$ | $\mathbf{0.616 \pm 0.015}$ |
| Linear (Laplacian+spectrogram) | $\mathbf{0.611 \pm 0.003}$ | $\mathbf{0.904 \pm 0.012}$ | $\mathbf{0.889 \pm 0.018}$ | $0.611 \pm 0.012$ |
| BrainBERT (untrained, frozen) | $0.552 \pm 0.003$ | $0.722 \pm 0.032$ | $0.609 \pm 0.016$ | $0.526 \pm 0.005$ |
| BrainBERT (frozen) | $0.557 \pm 0.003$ | $0.738 \pm 0.032$ | $0.631 \pm 0.018$ | $0.535 \pm 0.006$ |
| PopulationTransformer | $0.562 \pm 0.005$ | $0.760 \pm 0.034$ | $0.711 \pm 0.032$ | $0.561 \pm 0.014$ |

| Model | Delta Volume | Voice Pitch | Word Position | Inter-word Gap |
|---|---|---|---|---|
| Linear (voltage) | $0.596 \pm 0.005$ | $0.515 \pm 0.003$ | $0.642 \pm 0.016$ | $0.537 \pm 0.006$ |
| Linear (spectrogram) | $0.604 \pm 0.014$ | $0.530 \pm 0.005$ | $0.632 \pm 0.022$ | $0.532 \pm 0.010$ |
| Linear (Laplacian+spectrogram) | $0.630 \pm 0.014$ | $\mathbf{0.539 \pm 0.008}$ | $\mathbf{0.681 \pm 0.019}$ | $\mathbf{0.551 \pm 0.008}$ |
| BrainBERT (untrained, frozen) | $0.585 \pm 0.011$ | $0.503 \pm 0.002$ | $0.631 \pm 0.022$ | $0.525 \pm 0.010$ |
| BrainBERT (frozen) | $0.585 \pm 0.011$ | $0.507 \pm 0.003$ | $0.634 \pm 0.022$ | $0.529 \pm 0.009$ |
| PopulationTransformer | $\mathbf{0.632 \pm 0.025}$ | $0.517 \pm 0.008$ | $0.572 \pm 0.026$ | $0.526 \pm 0.012$ |

| Model | GPT-2 Surprisal | Head Word Position | Part of Speech | Word Length |
|---|---|---|---|---|
| Linear (voltage) | $0.529 \pm 0.006$ | $0.537 \pm 0.005$ | $0.530 \pm 0.005$ | $0.536 \pm 0.008$ |
| Linear (spectrogram) | $0.535 \pm 0.006$ | $0.557 \pm 0.011$ | $0.531 \pm 0.007$ | $0.537 \pm 0.009$ |
| Linear (Laplacian+spectrogram) | $0.547 \pm 0.007$ | $\mathbf{0.580 \pm 0.009}$ | $\mathbf{0.549 \pm 0.007}$ | $\mathbf{0.571 \pm 0.007}$ |
| BrainBERT (untrained, frozen) | $0.534 \pm 0.008$ | $0.579 \pm 0.015$ | $0.515 \pm 0.005$ | $0.541 \pm 0.010$ |
| BrainBERT (frozen) | $0.534 \pm 0.008$ | $0.577 \pm 0.015$ | $0.517 \pm 0.006$ | $0.540 \pm 0.009$ |
| PopulationTransformer | $\mathbf{0.558 \pm 0.014}$ | $0.523 \pm 0.006$ | $0.505 \pm 0.006$ | $0.526 \pm 0.006$ |

| Model | Global Optical Flow | Local Optical Flow | Frame Brightness | Number of Faces |
|---|---|---|---|---|
| Linear (voltage) | $0.510 \pm 0.002$ | $0.506 \pm 0.002$ | $0.504 \pm 0.004$ | $0.503 \pm 0.002$ |
| Linear (spectrogram) | $0.538 \pm 0.006$ | $0.534 \pm 0.008$ | $\mathbf{0.523 \pm 0.007}$ | $0.512 \pm 0.003$ |
| Linear (Laplacian+spectrogram) | $\mathbf{0.549 \pm 0.006}$ | $\mathbf{0.539 \pm 0.006}$ | $0.514 \pm 0.008$ | $\mathbf{0.516 \pm 0.006}$ |
| BrainBERT (untrained, frozen) | $0.509 \pm 0.004$ | $0.508 \pm 0.003$ | $0.499 \pm 0.002$ | $0.497 \pm 0.003$ |
| BrainBERT (frozen) | $0.510 \pm 0.003$ | $0.512 \pm 0.003$ | $0.506 \pm 0.006$ | $0.499 \pm 0.003$ |
| PopulationTransformer | $0.518 \pm 0.010$ | $0.516 \pm 0.012$ | $0.509 \pm 0.013$ | $0.497 \pm 0.008$ |

Supplementary Table 11: **Performance comparison across multi-class tasks (mean $\pm$ SEM) for within-session split.** Best performing model for each task is shown in bold.

| Model | Overall | Sentence Onset | Speech | Volume |
|---|---|---|---|---|
| Linear (voltage) | $0.575 \pm 0.003$ | $0.795 \pm 0.021$ | $0.656 \pm 0.022$ | $0.539 \pm 0.008$ |
| Linear (spectrogram) | $0.593 \pm 0.004$ | $0.851 \pm 0.025$ | $0.825 \pm 0.028$ | $\mathbf{0.615 \pm 0.022}$ |
| Linear (Laplacian+spectrogram) | $\mathbf{0.617 \pm 0.003}$ | $\mathbf{0.891 \pm 0.018}$ | $\mathbf{0.883 \pm 0.018}$ | $0.612 \pm 0.019$ |
| BrainBERT (untrained, frozen) | $0.560 \pm 0.003$ | $0.752 \pm 0.029$ | $0.598 \pm 0.021$ | $0.529 \pm 0.006$ |
| BrainBERT (frozen) | $0.560 \pm 0.003$ | $0.753 \pm 0.029$ | $0.606 \pm 0.022$ | $0.530 \pm 0.006$ |
| PopulationTransformer | $0.546 \pm 0.005$ | $0.725 \pm 0.043$ | $0.687 \pm 0.034$ | $0.561 \pm 0.016$ |

| Model | Delta Volume | Voice Pitch | Word Position | Inter-word Gap |
|---|---|---|---|---|
| Linear (voltage) | $0.623 \pm 0.011$ | $0.520 \pm 0.003$ | $0.696 \pm 0.016$ | $0.546 \pm 0.007$ |
| Linear (spectrogram) | $0.609 \pm 0.017$ | $0.532 \pm 0.005$ | $0.643 \pm 0.023$ | $0.542 \pm 0.012$ |
| Linear (Laplacian+spectrogram) | $\mathbf{0.640 \pm 0.017}$ | $\mathbf{0.541 \pm 0.007}$ | $\mathbf{0.712 \pm 0.022}$ | $\mathbf{0.558 \pm 0.008}$ |
| BrainBERT (untrained, frozen) | $0.599 \pm 0.012$ | $0.516 \pm 0.005$ | $0.640 \pm 0.021$ | $0.547 \pm 0.010$ |
| BrainBERT (frozen) | $0.599 \pm 0.013$ | $0.513 \pm 0.005$ | $0.641 \pm 0.022$ | $0.543 \pm 0.012$ |
| PopulationTransformer | $0.602 \pm 0.027$ | $0.508 \pm 0.011$ | $0.542 \pm 0.020$ | $0.517 \pm 0.014$ |

| Model | GPT-2 Surprisal | Head Word Position | Part of Speech | Word Length |
|---|---|---|---|---|
| Linear (voltage) | $0.539 \pm 0.006$ | $0.570 \pm 0.008$ | $0.537 \pm 0.005$ | $0.562 \pm 0.007$ |
| Linear (spectrogram) | $0.526 \pm 0.009$ | $0.565 \pm 0.012$ | $0.528 \pm 0.008$ | $0.538 \pm 0.010$ |
| Linear (Laplacian+spectrogram) | $\mathbf{0.555 \pm 0.008}$ | $\mathbf{0.602 \pm 0.012}$ | $\mathbf{0.551 \pm 0.008}$ | $\mathbf{0.569 \pm 0.010}$ |
| BrainBERT (untrained, frozen) | $0.536 \pm 0.007$ | $0.581 \pm 0.011$ | $0.524 \pm 0.004$ | $0.547 \pm 0.010$ |
| BrainBERT (frozen) | $0.537 \pm 0.008$ | $0.583 \pm 0.012$ | $0.522 \pm 0.004$ | $0.549 \pm 0.011$ |
| PopulationTransformer | $0.516 \pm 0.011$ | $0.511 \pm 0.004$ | $0.499 \pm 0.008$ | $0.507 \pm 0.009$ |

| Model | Global Optical Flow | Local Optical Flow | Frame Brightness | Number of Faces |
|---|---|---|---|---|
| Linear (voltage) | $0.526 \pm 0.006$ | $0.521 \pm 0.003$ | $0.500 \pm 0.008$ | $0.498 \pm 0.005$ |
| Linear (spectrogram) | $0.552 \pm 0.008$ | $0.539 \pm 0.012$ | $\mathbf{0.518 \pm 0.010}$ | $0.507 \pm 0.004$ |
| Linear (Laplacian+spectrogram) | $\mathbf{0.564 \pm 0.009}$ | $\mathbf{0.552 \pm 0.010}$ | $0.513 \pm 0.013$ | $\mathbf{0.514 \pm 0.007}$ |
| BrainBERT (untrained, frozen) | $0.514 \pm 0.005$ | $0.514 \pm 0.004$ | $0.503 \pm 0.003$ | $0.505 \pm 0.004$ |
| BrainBERT (frozen) | $0.511 \pm 0.005$ | $0.513 \pm 0.006$ | $0.501 \pm 0.005$ | $0.503 \pm 0.004$ |
| PopulationTransformer | $0.519 \pm 0.010$ | $0.516 \pm 0.010$ | $0.482 \pm 0.016$ | $0.498 \pm 0.010$ |

Supplementary Table 12: **Performance comparison across multi-class tasks (mean $\pm$ SEM) for cross-session split.** Best performing model for each task is shown in bold.

