# OpenReview forum: "Neuroprobe: Evaluating Intracranial Brain Responses to Naturalistic Stimuli"
_ICLR.cc/2026/Conference — Submitted to ICLR 2026_

### Official Review · Reviewer_8FJK · 2025-10-28

**Soundness:** 3
**Presentation:** 3
**Contribution:** 3
**Rating:** 8
**Confidence:** 3

**Summary:**

This paper introduces **Neuroprobe**, a multitask iEEG decoding benchmark (15 tasks spanning audio/vision/language) built on BrainTreebank, with standardized **within-session / cross-session / cross-subject** splits and AUROC as the primary metric. It establishes linear baselines and compares “frozen” foundation-model representations (BrainBERT, PopulationTransformer).

**Strengths:**

1. **High potential community impact.** Well-structured benchmark with clear splits, standardized electrodes per subject, and a plan for a public leaderboard, enabling fair and reproducible model comparison on diverse naturalistic tasks.

2. **Comprehensive evaluation.** The benchmark spans **15 decoding tasks** across auditory, visual, and linguistic domains, providing broad support for probing multimodal neural processing and enabling cross-modal insights under naturalistic stimuli.

3. **Strong and Transparent Baselines.** The benchmark presents some baseline models (from raw-signal linear decoders to frozen foundation-model representations) , which provide honest performance references that will help guide future progress.

4. **Insightful Spatial and Temporal Analyses**. Figs 4–7 effectively demonstrate how Neuroprobe enables neuroscientific discoveries, including modality-specific cortical organization and the temporal evolution of linguistic and perceptual feature processing.

**Weaknesses:**

1. **Task overview missing in the main text.** A schematic grouping the 15 tasks by modality/subtasks would improve immediate understanding.

2. **Foundation models only used as frozen encoders**, without justification or discussion on whether fine-tuning changes the ordering.

3. **Model size & compute not reported** in Fig. 3 & Fig. 7, limiting efficiency comparison.

4. The **counter-intuitive observation** that linear ≥ foundation models is not discussed in the paper.

**Questions:**

1. Please **add a compact visual** summarizing all 15 tasks (audio/language/vision groups and label generation).

2. Why **freeze foundation models** instead of fine-tuning? Were they pretrained on **BrainTreebank**? Any **subject or task overlap**? A brief description of each model in the main text would help.

3. Please **report parameters and size** for linear vs. foundation models in Fig. 3 & Fig. 7.

4. Please discuss **why** the tuned linear baseline outperforms foundation models—e.g., representation alignment, SNR improvements from Laplacian referencing , etc.

---

> ### Author Response · Authors · 2025-11-24
>
> Thank you for taking the time to read our paper and for your helpful suggestions! We address your comments and questions below:
>
> > add a compact visual summarizing all 15 tasks
>
> - Thanks for the suggestion. We adapt the handy overview figure and caption from the original BrainTreebank paper [1] and include it as supplementary figure 13 for reference.
>
> > Why freeze foundation models instead of fine-tuning? Were they pretrained on BrainTreebank? Any subject or task overlap?
> - We run decoding on frozen BrainBERT embeddings because running the entire fine-tuning suite across all tasks and time-bins on the single-electrode level is prohibitively computationally expensive for all >15,000 electrodes. BrainBERT and PopulationTransformer were trained on the same subjects used in Neuroprobe.
>
> > Please report parameters and size for linear vs. foundation models
> - We have added the model sizes in appendix in supplementary table 9.
>
> > Please discuss why the tuned linear baseline outperforms foundation models
> - This result may seem surprising, but makes more sense when you consider that pre-training for the foundation models wasn’t done with the Neuroprobe preprocessing and tasks in mind. We include results from BrainBERT and PopulationTransformer to show what is immediately possible for any user with access to the current off-the-shelf weights, as they are publicly available, but we do not expect these results to be representative of the ceiling for foundation model performance. In fact, we have a few experiments (see supplementary figure 15) that show that with a little bit of preprocessing optimization the decoding results change greatly. For example, it seems that the normalizing scheme that BrainBERT and PopulationTransformer use during pre-training is sub-optimal (see supplementary figure 16). This suggests that a few pre-training adjustments could lead to much stronger performance. This is one of the benefits of Neuroprobe: highlighting where foundation models can be tweaked.
>
> [1] Wang, Christopher, et al. "Brain treebank: Large-scale intracranial recordings from naturalistic language stimuli." Advances in Neural Information Processing Systems 37 (2024): 96505-96540.

---

### Official Review · Reviewer_kf5t · 2025-10-28

**Soundness:** 2
**Presentation:** 2
**Contribution:** 2
**Rating:** 4
**Confidence:** 4

**Summary:**

This paper introduces Neuroprobe, a benchmark suite for evaluating machine learning models on naturalistic stimuli decoding tasks to understand the neural computation mechanism of multi-modal language processing from human intracranial EEG (iEEG) data. Neuroprobe builds upon the BrainTreebank dataset, offering 15 well-defined binary classification tasks, standardized evaluation splits (within-session, cross-session, and cross-subject), and a leaderboard to track progress in iEEG foundation models. Through extensive analyses across spatial and temporal domains, the authors explore the flow of information in the brain and present comprehensive baseline comparisons (including linear models and state-of-the-art pretrained models).

**Strengths:**

1. Neuroprobe’s benchmark is carefully constructed from the movie tasks in Brain Treebank [1], featuring pre-defined tasks, rigorously constructed train/test splits, and standardized evaluation rules to evaluate previous baselines [2,3].

2. The authors conduct in-depth spatial and temporal analyses (based on linear decoding), using Neuroprobe to visualize how language, audio, and visual features are represented and evolve in the brain, where the analysis methods are widely adopted in neuroscience research. The discussion is substantively grounded, providing interpretable neuroscientific insights, such as the mapping of features to specific brain regions and the timing of language/visual processing.

3. The manuscript thoroughly documents the experimental setup and choices, making these tools (including open-source code and a leaderboard) publicly available to ensure reproducibility and community usability.

**Weaknesses:**

1. The motivation for expanding the original 4 decoding tasks (i.e., pitch, volume, sentence onset, and speech/non-speech in BrainBERT [2]) to 15 tasks is unclear. It’s unclear whether these manually defined concepts (e.g., number of faces, global optical flow) align with the neural encoding mechanism [4-6]. Could the author provide some neuroscience literature to ensure the neuroscientific significance of these decoding tasks?

2. The linear decoding results differ significantly from the results in [2] (e.g., 0.90 v.s. 0.60 on sentence onset), which greatly challenges the effectiveness of current iEEG foundation models. This work is similar to DLinear [7] in the time series forecasting field. If this work aims to challenge the representation learning of current iEEG foundation models (as the evaluation focuses on linear probing), more baselines [8-10] are needed for evaluation.

Overall, I appreciate the efforts in benchmark formulation and running so many baselines, but I am not satisfied with the novelty of the method.

**References**:

[1] Wang C, Yaari A, Singh A, et al. Brain treebank: Large-scale intracranial recordings from naturalistic language stimuli[J]. Advances in Neural Information Processing Systems, 2024, 37: 96505-96540.

[2] Wang C, Subramaniam V, Yaari A U, et al. BrainBERT: Self-supervised representation learning for intracranial recordings[J]. arXiv preprint arXiv:2302.14367, 2023.

[3] Chau G, Wang C, Talukder S, et al. Population transformer: Learning population-level representations of neural activity[J]. ArXiv, 2025: arXiv: 2406.03044 v4.

[4] Mesgarani N, Cheung C, Johnson K, et al. Phonetic feature encoding in human superior temporal gyrus[J]. Science, 2014, 343(6174): 1006-1010.

[5] Luo A F, Henderson M M, Tarr M J, et al. Brainscuba: Fine-grained natural language captions of visual cortex selectivity[J]. arXiv preprint arXiv:2310.04420, 2023.

[6] Bouchard K E, Mesgarani N, Johnson K, et al. Functional organization of human sensorimotor cortex for speech articulation[J]. Nature, 2013, 495(7441): 327-332.

[7] Zeng A, Chen M, Zhang L, et al. Are transformers effective for time series forecasting?[C]//Proceedings of the AAAI conference on artificial intelligence. 2023, 37(9): 11121-11128.

[8] Jiang W B, Zhao L M, Lu B L. Large brain model for learning generic representations with tremendous EEG data in BCI[J]. arXiv preprint arXiv:2405.18765, 2024.

[9] Wang J, Zhao S, Luo Z, et al. Cbramod: A criss-cross brain foundation model for eeg decoding[J]. arXiv preprint arXiv:2412.07236, 2024.

[10] Zhang D, Yuan Z, Yang Y, et al. Brant: Foundation model for intracranial neural signal[J]. Advances in Neural Information Processing Systems, 2023, 36: 26304-26321.

**Questions:**

See the above concerns.

---

> ### Author Response · Authors · 2025-11-25
>
> Thanks for taking the time to review our work and provide helpful feedback. We address your comments and questions below:
>
> > The motivation for expanding the original 4 decoding tasks … is unclear. It’s unclear whether these manually defined concepts (e.g., number of faces, global optical flow) align with the neural encoding mechanism
> - Thanks for the question. This is a good line of thought to clarify. In building Neuroprobe, we do not seek to propose a particular encoding mechanism. Whether or not the concepts we define align with neural encoding mechanisms is an open question! And answering this question is something that a benchmark like Neuroprobe can help enable. We argue that the first step in proposing an encoding mechanism, such as in the papers you cited, is determining whether that information is even decodable in the first place. The purpose of Neuroprobe is to make it easy to run these types of decoding experiments.
> - To clarify our reasoning behind proposing such a wide variety of tasks: probing for different aspects of multi-modal language processing is a line of research that has precedent in neuroscience [11, 12]. It’s well recognized that during language processing, the human brain integrates information from multiple channels simultaneously: visual, auditory, and language [13]. Neuroprobe includes a large variety of tasks to enable study of this.
>
> > The linear decoding results differ significantly from the results in [2] (e.g., 0.90 v.s. 0.60 on sentence onset), which greatly challenges the effectiveness of current iEEG foundation models.
> - Note that the BrainBERT results that we report are for the frozen off-the-shelf embeddings. The results we get are in line with Table 2 of [2] (e.g., 0.66 vs 0.60 on sentence onset). We will make sure to clarify this in the caption. There are a few other subtle differences that should be noted. In Neuroprobe, we take care to split our train/test/val based on time, rather than uniformly at random. This prevents models from using autocorrelation in the recording as a signal, and makes the task more difficult than the evaluation reported in [2]. Finally, we should point out that the BrainBERT embeddings are actually aggregated across channels before decoding happens, which doesn’t happen in the evaluation of [2]. We have reworked the caption of Figure 7 so that these differences are emphasized.
> - It was not our intention at all to challenge the effectiveness of current foundation models. We include results from BrainBERT and PopulationTransformer to show what is immediately possible for any user with access to the current off-the-shelf weights, as they are publicly available, but we do not expect these results to be representative of the ceiling for foundation model performance. In fact, we added a few experiments that show that with a little bit of preprocessing optimization the decoding results change greatly. For example, it seems that the normalizing scheme that BrainBERT and PopulationTransformer use during pre-training is sub-optimal (see supplementary figure 16). This suggests that pre-training adjustments could lead to much stronger performance. This is one of the benefits of Neuroprobe: highlighting where foundation models can be tweaked.
>
> [11] Subramaniam V, Conwell C, Wang C, Kreiman G, Katz B, Cases I, Barbu A. Revealing Vision-Language Integration in the Brain with Multimodal Networks. ArXiv [Preprint]. 2024 Jun 20:arXiv:2406.14481v1. PMID: 38947929; PMCID: PMC11213144.
>
> [12] Zhang Y, Kim JH, Brang D, Liu Z. Naturalistic Stimuli: A Paradigm for Multi-Scale Functional Characterization of the Human Brain. Curr Opin Biomed Eng. 2021 Sep;19:100298. doi: 10.1016/j.cobme.2021.100298. Epub 2021 Jun 2. PMID: 34423178; PMCID: PMC8376216.
>
> [13] Vigliocco G, Perniss P, Vinson D. Language as a multimodal phenomenon: implications for language learning, processing and evolution. Philos Trans R Soc Lond B Biol Sci. 2014 Sep 19;369(1651):20130292. doi: 10.1098/rstb.2013.0292. PMID: 25092660; PMCID: PMC4123671.

---

### Official Review · Reviewer_Sm6F · 2025-10-30

**Soundness:** 2
**Presentation:** 3
**Contribution:** 3
**Rating:** 4
**Confidence:** 4

**Summary:**

This paper introduces Neuroprobe, a benchmark built upon the BrainTreebank iEEG dataset designed to evaluate brain decoding models under naturalistic stimuli. Neuroprobe defines 15 classification tasks spanning audio, vision, and language domains, offering standardized data splits (within-session, cross-session, cross-subject) and carefully controlled baselines. Using this resource, the authors analyze the spatiotemporal organization of language and perceptual processes in the brain and compare a range of linear and state-of-the-art neural decoders. The paper includes extensive visualization and performance analysis, and aims to establish a public leaderboard to support progress in iEEG-based representation learning.

**Strengths:**

1. Relevance and Timeliness: The introduction of Neuroprobe as a standardized benchmark for iEEG decoding addresses a critical and widely recognized gap in the field. The lack of rigorous, standardized evaluation frameworks has hindered reproducible progress in iEEG foundation models. This benchmark, built on the publicly available BrainTreebank dataset, provides a much-needed common ground for comparing model performance and tracking advances.

2. Rigorous and Controlled Benchmarking: The paper thoughtfully defines three distinct evaluation splits (within-session, cross-session, cross-subject) that systematically address critical issues like temporal autocorrelation and test generalization across different levels of difficulty. The methodology for preventing data leakage (e.g., using contiguous blocks for cross-validation) is a particular strength, ensuring the integrity of the reported results.

3. Rich, Neuroscientifically Interpretable Analysis: A major strength of the work is the extensive analysis that translates model performance into neuroscientific insights. The spatial (Figure 4) and temporal (Figures 5, 6) visualizations of decodability provide a clear, data-driven picture of how and where different multimodal features are processed in the human brain. This dual utility—as both a machine learning benchmark and a tool for neuroscience discovery—significantly enhances the paper's impact.

**Weaknesses:**

1. Limited Scope of Model Benchmarking: While the authors benchmark a linear baseline and two foundation models (BrainBERT, PopulationTransformer), the evaluation lacks a broader range of modern deep learning architectures. For instance, non-foundation models like supervised CNN-Transformer or RNNs, which have shown strong performance in related neural signal decoding tasks [1-3] , are absent. A more diverse set of baselines would provide a clearer picture of the current state-of-the-art and the specific challenges posed by this benchmark.

2. Insufficient Analysis of Anatomical Variability Impact: The benchmark acknowledges the challenge of variable electrode placements across subjects but does not quantitatively analyze how this anatomical heterogeneity affects decoding performance, especially in the challenging cross-subject split. An ablation study examining performance relative to inter-subject electrode coverage overlap or a more detailed regional analysis would strengthen the conclusions about the spatial localization of decodable features.

3. Incomplete Contemporary iEEG Literature: The related work section provides a good overview of the benchmark landscape but could be more thoroughly integrated with recent advances in iEEG/ECoG deep learning. For example, several recent works focusing on deep learning for functional mapping, unsupervised representation learning, or subject-transfer in iEEG are highly relevant [4-6]. Engaging more directly with this literature would better contextualize Neuroprobe's contributions and suggest immediate avenues for future benchmarking by the community.

**References:**

[1] Willett, F. R., Kunz, E. M., Fan, C., Avansino, D. T., Wilson, G. H., Choi, E. Y., ... & Henderson, J. M. (2023). A high-performance speech neuroprosthesis. *Nature, 620*(7976), 1031-1036.

[2] Metzger, S. L., Littlejohn, K. T., Silva, A. B., Moses, D. A., Seaton, M. P., Wang, R., ... & Chang, E. F. (2023). A high-performance neuroprosthesis for speech decoding and avatar control. *Nature, 620*(7976), 1037-1046.

[3] Song, Y., Zheng, Q., Liu, B., & Gao, X. (2022). EEG conformer: Convolutional transformer for EEG decoding and visualization. *IEEE Transactions on Neural Systems and Rehabilitation Engineering, 31*, 710-719.

[4] Chen, X., Wang, R., Khalilian-Gourtani, A., Yu, L., Dugan, P., Friedman, D., ... & Flinker, A. (2024). A neural speech decoding framework leveraging deep learning and speech synthesis. *Nature Machine Intelligence, 6*(4), 467-480.

[5] Zheng, H., Wang, H., Jiang, W., Chen, Z., He, L., Lin, P., ... & Liu, Y. (2024). Du-IN: Discrete units-guided mask modeling for decoding speech from Intracranial Neural signals. *Advances in Neural Information Processing Systems, 37*, 79996-80033.

[6] Singh, A., Thomas, T., Li, J., Hickok, G., Pitkow, X., & Tandon, N. (2025). Transfer learning via distributed brain recordings enables reliable speech decoding. *Nature Communications, 16*(1), 8749.

**Questions:**

See Weaknesses.

**Details Of Ethics Concerns:**

No ethics concerns are apparent. The public datasets are used appropriately.

---

> ### Author Response · Authors · 2025-11-25
>
> Thanks for the helpful feedback. We appreciate your comments. Below, we address the questions raised in your review:
>
> > lacks a broader range of modern deep learning architectures.
>
> - We have now run more supervised baselines including a CNN and MLP, in addition to the linear baseline and foundation models from before. Results for these can be seen in supplementary figure 15. Of course, it’s always the case that even more baselines can be included, but we would like to emphasize that this paper is meant to serve more as a release of the data/benchmark. Our main goals with this project is to set the standards for reporting, and to gain some insight into the neuroscience analyses that are possible with the benchmark, i.e. Figures 4,5,6.
>
>
> > does not quantitatively analyze how this anatomical heterogeneity affects decoding performance, especially in the challenging cross-subject split … more detailed regional analysis
>
> - This is an interesting question. For the cross-subject split, we can quantify the region overlap between the test subjects and the train section as the intersection-over-union over the different Desikan-Killiany regions that are sampled for each subject. We have added a table (supplementary table 8) that shows the relationship between overlap and task performance. Since there are only 4 test subjects, we can not say anything conclusive, but there seems to be even a slight negative correlation for the present subjects.
>
> > a good overview of the benchmark landscape but could be more thoroughly integrated with recent advances in iEEG/ECoG deep learning…contextualize Neuroprobe's contributions
>
> - Thanks for the suggestion. We’ve added citations to those papers and a few others under related works as part of a discussion on modeling work that is being done with this sort of data. That said, we elected to keep our related works section mainly focused on prior benchmarking efforts, since our contribution is mainly a dataset/benchmark. When it comes to contextualizing Neuroprobe, our most significant contribution is providing a benchmark that uses (1) high-resolution iEEG data and (2) rich multi-modal stimuli. Everything we’ve found has been focused on a single modality or task, which is why we widely cite existing datasets to give evidence of this.

---

### Official Review · Reviewer_L6Xh · 2025-10-31

**Soundness:** 3
**Presentation:** 2
**Contribution:** 3
**Rating:** 6
**Confidence:** 4

**Summary:**

This work introduces what appears to be the first public benchmark that systematizes high-fidelity naturalistic iEEG into a multi-task, cross-session, and cross-subject evaluation suite. The design is clear, the implementation is practical, and the results are reproducible with code and a leaderboard.
This paper introduces a public benchmark that turns high-fidelity naturalistic iEEG into a coherent evaluation suite: it builds on BrainTreebank’s 40+ hours from 10 participants, defines 15 standardized decoding tasks spanning speech, vision, and language, and ships unified protocols, metrics, code, and a leaderboard.  It formalizes three splits—within-session, cross-session, and cross-subject—with safeguards against temporal leakage and a default focus on cross-session testing.

It lowers the barrier to entry with a Neuroprobe-Lite subset, a cap of 120 electrodes per subject, probe reuse for multiple re-referencing strategies, and a clear submission workflow. Finally, it benchmarks linear models, BrainBERT, and PopulationTransformer, and finds that Laplacian-re-referenced spectrograms with a simple linear model are surprisingly strong and already yield clear spatiotemporal decoding maps and processing order.

Several tasks are binarized and a few evaluation choices may limit external validity. The paper would be stronger if the main text added motivation for binarization, robustness checks across thresholds, and small ablations to show conclusions are stable.

**Strengths:**

Originality. First standardized benchmark from high-fidelity, natural iEEG. It turns movie-watching data into 15 clear decoding tasks across speech, vision, and language, and tests three settings: within-session, cross-session, and cross-subject.

Quality. Careful and reproducible setup. Splits prevent temporal leakage, inputs are fixed at one second, and a 120-electrode cap keeps things practical while allowing re-referencing. Baselines span simple linear models to pretrained transformers, with documented spectrogram choices and a public leaderboard.

Clarity. Easy to follow. Figures show how raw data become tasks, what each split measures, and how models perform across all 15 tasks. Labels and windowing are defined plainly, and the appendix table removes ambiguity.

Significance. Provides a reusable yardstick for iEEG foundation models. Simple Laplacian-based spectrograms with linear decoding already do very well and yield clear spatiotemporal maps of decodability, which will guide future methods and fair comparisons.

**Weaknesses:**

1 Many tasks are reduced to binary by percentile thresholds or one-vs-rest, which can distort difficulty and hide effects. Please justify the choice in the main text and add robustness checks: threshold sweeps, and where possible, regression or multi-class versions with AUROC/PR-AUC or R².


2 Authors cross-subject protocol trains only on Subject 2, Trial 4, then tests on others. This is a very narrow source distribution and may not generalize. Add alternatives: train on a different subject/session, a pooled multi-source train set, and report how rankings and absolute scores move.

3 For efficiency authors cap each subject at 120 electrodes and choose them starting with those that show the highest linear decoding performance. That can create circularity and make simple linear probes look stronger than they would under anatomy-only or random selection. Please clarify the selection is done using train-only data, and add sensitivity analyses: anatomy-driven, random, and performance-driven selection compared side by side.

4. Baseline parity and “why linear wins” are under-analyzed.
The strongest baseline is linear decoding on Laplacian-re-referenced spectrograms, helped by careful spectrogram sweeps. The pretrained models use different inputs and may be handicapped. To make the comparison fair and more informative, add runs where all models share the same pre-processing (e.g., Laplacian spectrograms), and a small ablation quantifying the contribution of re-referencing and spectrogram hyperparameters to linear gains.

**Questions:**

1 Are task windows anchored appropriately for each modality?
All evaluations cut a 1-second window after each word onset. That is natural for language, but visual tasks (e.g., optical flow, brightness) may not align to word timing.
Would it be fair to test modality-specific anchors (e.g., frame onsets or peak optical-flow times) and add a jitter-robustness check that shifts windows ±250–500 ms to confirm conclusions hold.

2 Does the Lite subset’s truncation introduce bias?
Neuroprobe-Lite keeps the first 3,500 annotations per task and selects subject–trial pairs that hit this limit. That “first-N” rule can skew content distributions and label balance.
 Please report how class proportions and movie segments differ before vs. after trimming, and consider stratified sampling across the full session to avoid temporal bias.

3 Is the cross-subject region mapping discarding informative data?
For cross-subject, authors average activity within atlas regions and only keep regions shared by both subjects. This simplifies alignment but may drop informative electrodes and change task difficulty. Please quantify how many regions are discarded per pair, show performance vs. fraction of overlapping regions, and compare with alternative alignments (e.g., functional correspondence or optimal transport).

4 How will the leaderboard guard against overfitting?
Submissions are unverified beyond formatting and receive a “reproducible” tag if a code link is provided. To curb test-set overfitting, consider a sequestered hidden test split, periodic refreshes, and mandatory config files with fixed seeds.

**Details Of Ethics Concerns:**

An ethics review is warranted for privacy and human-subjects risks: the dataset contains invasive iEEG from clinical patients, including minors, so the paper should spell out consent scope and IRB approvals, re-identification safeguards, data handling, and licensing. The authors should also commit to subgroup fairness reporting, tighten leaderboard governance to deter test overfitting, and publish a clear misuse policy.

---

> ### Author Response · Authors · 2025-11-25
> **Author Response (1/3)**
>
> Thanks for taking the time to read our paper and leaving a thorough review. The feedback was very helpful. We address your questions and comments below:
>
> > Many … binary by percentile thresholds or one-vs-rest, which can distort difficulty … justify the choice in the main text and add robustness checks: threshold sweeps, and where possible, regression or multi-class versions with AUROC/PR-AUC or R².
>
> - Thanks for this suggestion; we’ve modified the Neuroprobe codebase so now users can set `binary_tasks=False` in the `generate_splits` function to obtain multi-class versions of the tasks where relevant. The full multiclass task list can now be seen in Supplementary Table 8, and we clarify the multiclass Neuroprobe option in the main text of the paper. The main text still contains the default binary-task results, since they allow for easy aggregation and discussion of tasks as a group, but we will have multi-class results for the camera ready.
>
> > Authors cross-subject protocol trains only on Subject 2, Trial 4, then tests on others. … narrow source distribution ... Add alternatives: train on a different subject/session, a pooled multi-source train set
>
> - Originally, we selected only one subject due to computational considerations, and a desire to make the task easier for submitters to run. But to make a more complete picture of generalizability available, we have now added the option `include_all_train_subjects=True` in the `generate_splits` function. If this flag is passed, then the cross-subject split will train/test on all possible subject pairs in Neuroprobe. For now, we report the previous CrossSubject results, due to the time-cost of running all pairs, but will have the all pairs results at camera ready.
>
> > Does the Lite subset’s truncation introduce bias? Neuroprobe-Lite keeps the first 3,500 annotations per task and selects subject–trial pairs that hit this limit. That “first-N” rule can skew content distributions and label balance. Please report how class proportions and movie segments differ before vs. after trimming, and consider stratified sampling across the full session to avoid temporal bias.
>
> - Good question. We have clarified this in the main text: the first 3,500 annotations refer to the total across the classes, and class balance is maintained (so, for example, in a 2-class task, “taking the first 3500 annotations” means specifically “take the first 1750 samples of class 1 and the first 1750 samples of class 2”, which guarantees the class balance by design.). At the moment, we don’t stratify sampling across time, but in any case, all models will receive the same samples during training, so fair comparisons can still be made.
>
> >  For efficiency authors cap each subject at 120 electrodes and choose them starting with those that show the highest linear decoding performance. That can create circularity …  add sensitivity analyses: anatomy-driven, random, and performance-driven selection compared side by side.
>
> - Your point about circularity is well taken. We have now tested out a few different sets of electrodes. We add this plot as Supplementary Figure 12. In particular, we ran the linear decoding when the electrodes are selected by a performance-driven selection (default), sampled at random uniformly without replacement, and anatomy driven – we only select frontal and temporal electrodes. The performance driven selection results in only slightly better linear decodability, and even this is not the case across all tasks.

---

> ### Author Response · Authors · 2025-11-25
> **Author reponse (2/3)**
>
> > Baseline parity and “why linear wins” are under-analyzed. … The pretrained models use different inputs and may be handicapped. … add runs where all models share the same pre-processing (e.g., Laplacian spectrograms), and a small ablation quantifying the contribution of re-referencing and spectrogram hyperparameters to linear gains.
>
> - We include results from BrainBERT and PopulationTransformer to show what is immediately possible for any user with access to the current off-the-shelf weights, as they are publicly available, but we do not expect these results to be representative of the ceiling for foundation model performance. We will make sure to clarify that these results are meant to be read as such. While the focus of this specific paper is not on developing new pretraining and preprocessing methods, we suspect that part of the reason for lower performance is due to how BrainBERT and PopT select their context window length in pretraining. Specifically, those papers specifically train on 5-second fixed invariable time intervals, which might undermine their performance on Neuroprobe which gives only 1 second of input data (see [1]). It’s likely that with solid preprocessing optimization, the foundation models will perform better.
> - At your suggestion we investigated further and found that the specific way that the spectrograms were normalized in the original foundation models (BrainBERT and PopT) may also contribute. Specifically, the original models z-scored their inputs within the 5-second intervals which constitute the inputs, whereas in our Linear baseline, we z-scored the data across the whole (train) dataset. The BrainBERT z-scoring lost information about the relative value within this 5-second interval compared to the other 5-second intervals in the dataset. We have confirmed this intuition by running Neuroprobe evaluations on the Linear baseline which recapitulated the z-scoring procedure of BrainBERT and PopT, and indeed found a significantly decreased Neuroprobe performance (Supplementary Figure 16).
>
> [1] Patel, E., Yue, Y., & Chau, G. (2025). Learning time-scale invariant population-level neural representations. arXiv. https://arxiv.org/abs/2511.13022
>
> > Are task windows anchored appropriately for each modality? All evaluations cut a 1-second window after each word onset. That is natural for language, but visual tasks (e.g., optical flow, brightness) may not align to word timing. Would it be fair to test modality-specific anchors (e.g., frame onsets or peak optical-flow times) and add a jitter-robustness check that shifts windows ±250–500 ms to confirm conclusions hold.
> - The main interest of this benchmark is in decoding multi-modal processing of language, hence why word timings form the anchors. But you raise very interesting questions. We ran additional analyses, and we found that across tasks, the decodability peaks around 0.0-0.25 seconds after the word onset. We also ran a jitter-robustness check as you suggested. We found that across all tasks, the one-second interval can start anywhere from -0.375s to 0.125s relative to the word onset, and the decoding performance is roughly identical, and outside of this window the performance starts to visibly drop. The data now reported in the appendix (Supplementary Figure 14). We also added a reference to it in the main text of the paper, when describing the choice of the input data context window.
>
> > Is the cross-subject region mapping discarding informative data? For cross-subject, authors average activity within atlas regions and only keep regions shared by both subjects. … This simplifies alignment but may drop informative electrodes and change task difficulty. … quantify how many regions are discarded per pair, show performance vs. fraction of overlapping regions, and compare with alternative alignments
> - The average task performance and region overlap can now be seen in supplementary table 9. And yes, you are right, in the model we present, our region-mapping is likely discarding informative data. But please keep in mind, these are would-be flaws of our baseline model. The baselines we report are given just to provide a sense of what is possible with a reasonable decoding attempt. Indeed, we would encourage submitters to Neuroprobe to explore alternative alignments like the ones you mention!

---

> ### Author Response · Authors · 2025-11-25
> **Author response (3/3)**
>
> > How will the leaderboard guard against overfitting?...To curb test-set overfitting, consider a sequestered hidden test split, periodic refreshes, and mandatory config files with fixed seeds.
> - These are good suggestions. We will fix random seeds in the config and ask submitters to adhere to using those seeds. Creating a sequestered hidden test split isn’t possible at this time, as Neuroprobe is built on top of previously publicly released recordings. However, it’s worth noting that other datasets like CIFAR100 and ImageNet have been massively useful for the machine learning community as a source for benchmark tasks, even without containing a sequestered split.
>
> > An ethics review is warranted for privacy and human-subjects risks: the dataset contains invasive iEEG from clinical patients, including minors, so the paper should spell out consent scope and IRB approvals, re-identification safeguards, data handling, and licensing. The authors should also commit to subgroup fairness reporting, tighten leaderboard governance to deter test overfitting, and publish a clear misuse policy.
> - Just to be clear: this benchmark makes use of publicly available, anonymized data. All data was released in BrainTreebank [2], and the IRB for the data is discussed in that paper. We will add a statement about misuse to the website.
>
> [2] Wang, Christopher, et al. "Brain treebank: Large-scale intracranial recordings from naturalistic language stimuli." Advances in Neural Information Processing Systems 37 (2024): 96505-96540.

---

> ### Comment · Reviewer_L6Xh · 2025-11-26
>
> The authors actually ran targeted ablations to quantify the impact of the normalization strategy, rather than hand-waving. While they did not fully realize the ideal setting where all models share exactly the same inputs and preprocessing, the additional experiments convincingly show two key points: (1) the current foundation-model pipeline is suboptimal for this benchmark, and (2) part of the linear model’s advantage stems from a better-matched preprocessing pipeline rather than some inherent superiority of linear models.
>
> However, several core concerns remain insufficiently addressed.
> (1) Robustness of main conclusions beyond binary tasks. The central question was whether the main findings (e.g., performance rankings, spatial/temporal patterns) still hold under multiclass formulations and different threshold choices. The rebuttal only adds a code option and promises multiclass results for the camera-ready version, but does not present any concrete multiclass or threshold-sweep results now. This leaves the robustness of the core conclusions underexamined.
> (2) Bias in the Neuroprobe-Lite subset. The concern was that taking the first 3,500 annotations per task could introduce temporal and content bias, and that the authors should compare label/moment distributions before vs. after truncation or adopt stratified sampling. The rebuttal instead argues that all models see the same Lite subset, so comparisons between models are fair. This only addresses fairness across models, not whether the Lite subset is representative of the full dataset or whether conclusions from Lite generalize to the full distribution.

---

> > ### Author Response · Authors · 2025-12-03
> > **Response to last final two points**
> >
> > 1. The multi-class results can now be seen in Supplementary tables 11 and 12. See below for cross-session results. The same trends hold.
> > 2. I see. I now understand what you mean by “temporal stratification”. The good news is that we in fact do this stratification! The proof of this is in lines 221-222 of `neuroprobe/datasets.py`. You can see that we shuffle the indices across time before sub-setting them.
> >
> > # Performance Comparison - CrossSession (Neuroprobe MultiClass)
> >
> > Performance comparison across tasks (mean ± SEM). **Best performing model** for each task is shown in bold.
> >
> > | Model | Overall | Sentence Onset | Speech | Volume |
> > |-------|---------|---------|---------|---------|
> > | Linear (voltage) | 0.554 ± 0.002 | 0.728 ± 0.021 | 0.611 ± 0.014 | 0.530 ± 0.003 |
> > | Linear (spectrogram) | 0.593 ± 0.003 | 0.861 ± 0.016 | 0.849 ± 0.020 | **0.616 ± 0.015** |
> > | Linear (Laplacian+spectrogram) | **0.611 ± 0.003** | **0.904 ± 0.012** | **0.889 ± 0.018** | 0.611 ± 0.012 |
> > | BrainBERT (untrained, frozen) | 0.552 ± 0.003 | 0.722 ± 0.032 | 0.609 ± 0.016 | 0.526 ± 0.005 |
> > | BrainBERT (frozen) | 0.557 ± 0.003 | 0.738 ± 0.032 | 0.631 ± 0.018 | 0.535 ± 0.006 |
> > | PopulationTransformer | 0.562 ± 0.005 | 0.760 ± 0.034 | 0.711 ± 0.032 | 0.561 ± 0.014 |
> >
> > | Model | Delta Volume | Voice Pitch | Word Position | Inter-word Gap |
> > |-------|---------|---------|---------|---------|
> > | Linear (voltage) | 0.596 ± 0.005 | 0.515 ± 0.003 | 0.642 ± 0.016 | 0.537 ± 0.006 |
> > | Linear (spectrogram) | 0.604 ± 0.014 | 0.530 ± 0.005 | 0.632 ± 0.022 | 0.532 ± 0.010 |
> > | Linear (Laplacian+spectrogram) | 0.630 ± 0.014 | **0.539 ± 0.008** | **0.681 ± 0.019** | **0.551 ± 0.008** |
> > | BrainBERT (untrained, frozen) | 0.585 ± 0.011 | 0.503 ± 0.002 | 0.631 ± 0.022 | 0.525 ± 0.010 |
> > | BrainBERT (frozen) | 0.585 ± 0.011 | 0.507 ± 0.003 | 0.634 ± 0.022 | 0.529 ± 0.009 |
> > | PopulationTransformer | **0.632 ± 0.025** | 0.517 ± 0.008 | 0.572 ± 0.026 | 0.526 ± 0.012 |
> >
> > | Model | GPT-2 Surprisal | Head Word Position | Part of Speech | Word Length |
> > |-------|---------|---------|---------|---------|
> > | Linear (voltage) | 0.529 ± 0.006 | 0.537 ± 0.005 | 0.530 ± 0.005 | 0.536 ± 0.008 |
> > | Linear (spectrogram) | 0.535 ± 0.006 | 0.557 ± 0.011 | 0.531 ± 0.007 | 0.537 ± 0.009 |
> > | Linear (Laplacian+spectrogram) | 0.547 ± 0.007 | **0.580 ± 0.009** | **0.549 ± 0.007** | **0.571 ± 0.007** |
> > | BrainBERT (untrained, frozen) | 0.534 ± 0.008 | 0.579 ± 0.015 | 0.515 ± 0.005 | 0.541 ± 0.010 |
> > | BrainBERT (frozen) | 0.534 ± 0.008 | 0.577 ± 0.015 | 0.517 ± 0.006 | 0.540 ± 0.009 |
> > | PopulationTransformer | **0.558 ± 0.014** | 0.523 ± 0.006 | 0.505 ± 0.006 | 0.526 ± 0.006 |
> >
> > | Model | Global Optical Flow | Local Optical Flow | Frame Brightness | Number of Faces |
> > |-------|---------|---------|---------|---------|
> > | Linear (voltage) | 0.510 ± 0.002 | 0.506 ± 0.002 | 0.504 ± 0.004 | 0.503 ± 0.002 |
> > | Linear (spectrogram) | 0.538 ± 0.006 | 0.534 ± 0.008 | **0.523 ± 0.007** | 0.512 ± 0.003 |
> > | Linear (Laplacian+spectrogram) | **0.549 ± 0.006** | **0.539 ± 0.006** | 0.514 ± 0.008 | **0.516 ± 0.006** |
> > | BrainBERT (untrained, frozen) | 0.509 ± 0.004 | 0.508 ± 0.003 | 0.499 ± 0.002 | 0.497 ± 0.003 |
> > | BrainBERT (frozen) | 0.510 ± 0.003 | 0.512 ± 0.003 | 0.506 ± 0.006 | 0.499 ± 0.003 |
> > | PopulationTransformer | 0.518 ± 0.010 | 0.516 ± 0.012 | 0.509 ± 0.013 | 0.497 ± 0.008 |

---

### Author Response · Authors · 2025-12-03
**Note to AC and all reviewers**

We thank the reviewers for reading our paper and engaging in discussion. Reviewers noted that Neuroprobe:

- Has the ability to “enable neuroscientific discoveries” **8FJK** and has “dual utility as both a machine learning benchmark and a tool for neuroscience discovery” **Sm6F**
- Has a “Careful and reproducible setup” and “provides a reusable yardstick for iEEG foundation model” **L6Xh**
- Presents “comprehensive baseline comparisons“ **kf5t**

To aid the AC in parsing discussion, we break our response to reviewers into 3 categories:

1. **Fulfilling straightforward requests**: Reviewer 8FJK asked for tests to show that our results are robust to electrode selection, label anchoring, and multi-class labeling. These can now be seen below and in supplementary figures 12, 14, and supplementary tables 11, 12 respectively. Reviewer Sm6F requested additional baselines, which we include (see below and in supplementary figure 15). Reviewers L6Xh and Sm6F wanted to know the relationship between region overlap between train and test subjects and performance, which we compute for supplementary figure 9\.
2. **Clarifying our position**: Reviewer 8FJK pointed out several ways in which our baseline model could be improved. Reviewer Sm6F asked us to engage more with the modeling literature. Reviewer kf5t asked for a proposed neural mechanism behind our benchmark tasks. To address these points, we clarify broadly that our main contribution is the creation of a benchmark and evidence that it can yield neuroscientific insights. We do not claim that our preliminary baselines are SOTA (to 8FJK’s point), nor do we attempt to propose a novel SOTA architecture (to Sm6F’s point). And while we show that Neuroprobe is very effective at probing the decodability of tasks, we do not posit any specific mechanism by which the brain encodes information (to kf5t’s point)
3. **Explaining interpretation of benchmark results**: Reviewers L6Xh, Sm6F, and 8FJK wondered why BrainBERT and PopulationTransformer give low performance on some tasks. This result may seem surprising, but makes more sense when you consider that pre-training for the foundation models wasn’t done with the Neuroprobe preprocessing and tasks in mind. We include results from BrainBERT and PopulationTransformer simply to show what is immediately possible for any user with access to the current off-the-shelf weights, as they are publicly available, but we do not expect these results to be representative of the ceiling for foundation model performance. We clarify in the text that these results are meant to be taken as such. We ran a few experiments (see supplementary figure 15) that show that with a little bit of preprocessing optimization, the decoding results change greatly. For example, it seems that the normalizing scheme used by BrainBERT and PopulationTransformer during pre-training is sub-optimal (see supplementary figure 16). This suggests that a few pre-training adjustments could lead to much stronger performance. In fact, this is one of the benefits of Neuroprobe: highlighting where foundation models can be tweaked.

**To summarize**: We thank the reviewers for their feedback. We clarify that rather than present a SOTA model, our work mainly serves to introduce the Neuroprobe benchmark and demonstrate how neuroscience insights can be derived from it. From our reading of the initial reviews, it seems that reviewers agree that Neuroprobe is effective and useful for the community for these purposes.

### Flag for ethics review
Reviewer L6Xh raised a flag for ethics review based on the fact that the data is collected via an invasive surgery on minors. Just to be clear: this benchmark makes use of publicly available, anonymized data. All recordings were released in BrainTreebank [1], and the IRB for the data is discussed in that paper. We will add a statement about misuse to the website.

[1] Wang, Christopher, et al. "Brain treebank: Large-scale intracranial recordings from naturalistic language stimuli." Advances in Neural Information Processing Systems 37 (2024): 96505-96540.

---

> ### Author Response · Authors · 2025-12-03
> **Supporting tables (1/3)**
>
> See below for some additional results we computed for this rebuttal. The complete set can be seen in the appendix, colored blue:
>
> In addition to BrainBERT and the PopulationTransformer, we now report additional baseline results, as requested by **Sm6F**, for a CNN and MLP:
>
> | task                | CNN       | MLP       |
> |:--------------------|:----------|:----------|
> | Delta Volume        | 0.76±0.09 | 0.75±0.1  |
> | Frame Brightness    | 0.53±0.07 | 0.51±0.07 |
> | GPT-2 Surprisal     | 0.6±0.06  | 0.61±0.06 |
> | Global Optical Flow | 0.64±0.06 | 0.63±0.05 |
> | Head Word Position  | 0.61±0.05 | 0.61±0.05 |
> | Inter-word Gap      | 0.58±0.06 | 0.6±0.05  |
> | Local Optical Flow  | 0.63±0.05 | 0.62±0.06 |
> | Number of Faces     | 0.54±0.04 | 0.52±0.06 |
> | Part of Speech      | 0.62±0.08 | 0.59±0.06 |
> | Sentence Onset      | 0.89±0.05 | 0.89±0.07 |
> | Speech              | 0.9±0.05  | 0.88±0.07 |
> | Voice Pitch         | 0.61±0.07 | 0.59±0.06 |
> | Volume              | 0.75±0.14 | 0.73±0.12 |
> | Word Length         | 0.64±0.07 | 0.61±0.07 |
> | Word Position       | 0.73±0.1  | 0.73±0.11 |
>
> We compute the effect of electrode selection, as requested by **L6Xh**, and show that it does not significantly affect results:
>
> | task                | Linear: Anatomical random seed 0   | Linear: Random seed 0   | Linear: Vanilla   |
> |:--------------------|:-----------------------------------|:------------------------|:------------------|
> | Delta Volume        | 0.73±0.1                           | 0.72±0.09               | 0.76±0.1          |
> | Frame Brightness    | 0.56±0.11                          | 0.56±0.09               | 0.53±0.1          |
> | GPT-2 Surprisal     | 0.6±0.06                           | 0.59±0.06               | 0.61±0.06         |
> | Global Optical Flow | 0.62±0.06                          | 0.61±0.06               | 0.63±0.05         |
> | Head Word Position  | 0.59±0.04                          | 0.59±0.04               | 0.6±0.04          |
> | Inter-word Gap      | 0.6±0.04                           | 0.58±0.06               | 0.61±0.05         |
> | Local Optical Flow  | 0.6±0.07                           | 0.6±0.08                | 0.61±0.06         |
> | Number of Faces     | 0.53±0.05                          | 0.54±0.05               | 0.52±0.06         |
> | Part of Speech      | 0.6±0.05                           | 0.59±0.06               | 0.6±0.06          |
> | Sentence Onset      | 0.86±0.09                          | 0.84±0.1                | 0.89±0.06         |
> | Speech              | 0.86±0.1                           | 0.84±0.08               | 0.88±0.06         |
> | Voice Pitch         | 0.57±0.06                          | 0.57±0.07               | 0.58±0.05         |
> | Volume              | 0.71±0.13                          | 0.74±0.12               | 0.72±0.13         |
> | Word Length         | 0.6±0.07                           | 0.6±0.07                | 0.62±0.06         |
> | Word Position       | 0.71±0.12                          | 0.71±0.1                | 0.74±0.11         |
>
> We compute performance as a function of region overlap with the test subject, as requested by **L6Xh** and **Sm6F** :
>
> |   avg\_test |       IOU | subject   |
> |-----------:|----------:|:----------|
> |   0.578849 | 0.0416667 | sub\_1     |
> |   0.51713  | 0.0714286 | sub\_10    |
> |   0.540566 | 0.371429  | sub\_3     |
> |   0.501422 | 0.47619   | sub\_7     |

---

> ### Author Response · Authors · 2025-12-03
> **Supporting tables (2/3)**
>
> # Performance Comparison - CrossSession (Neuroprobe MultiClass)
>
> Performance comparison across tasks (mean ± SEM). **Best performing model** for each task is shown in bold.
>
> | Model | Overall | Sentence Onset | Speech | Volume |
> |-------|---------|---------|---------|---------|
> | Linear (voltage) | 0.554 ± 0.002 | 0.728 ± 0.021 | 0.611 ± 0.014 | 0.530 ± 0.003 |
> | Linear (spectrogram) | 0.593 ± 0.003 | 0.861 ± 0.016 | 0.849 ± 0.020 | **0.616 ± 0.015** |
> | Linear (Laplacian+spectrogram) | **0.611 ± 0.003** | **0.904 ± 0.012** | **0.889 ± 0.018** | 0.611 ± 0.012 |
> | BrainBERT (untrained, frozen) | 0.552 ± 0.003 | 0.722 ± 0.032 | 0.609 ± 0.016 | 0.526 ± 0.005 |
> | BrainBERT (frozen) | 0.557 ± 0.003 | 0.738 ± 0.032 | 0.631 ± 0.018 | 0.535 ± 0.006 |
> | PopulationTransformer | 0.562 ± 0.005 | 0.760 ± 0.034 | 0.711 ± 0.032 | 0.561 ± 0.014 |
>
> | Model | Delta Volume | Voice Pitch | Word Position | Inter-word Gap |
> |-------|---------|---------|---------|---------|
> | Linear (voltage) | 0.596 ± 0.005 | 0.515 ± 0.003 | 0.642 ± 0.016 | 0.537 ± 0.006 |
> | Linear (spectrogram) | 0.604 ± 0.014 | 0.530 ± 0.005 | 0.632 ± 0.022 | 0.532 ± 0.010 |
> | Linear (Laplacian+spectrogram) | 0.630 ± 0.014 | **0.539 ± 0.008** | **0.681 ± 0.019** | **0.551 ± 0.008** |
> | BrainBERT (untrained, frozen) | 0.585 ± 0.011 | 0.503 ± 0.002 | 0.631 ± 0.022 | 0.525 ± 0.010 |
> | BrainBERT (frozen) | 0.585 ± 0.011 | 0.507 ± 0.003 | 0.634 ± 0.022 | 0.529 ± 0.009 |
> | PopulationTransformer | **0.632 ± 0.025** | 0.517 ± 0.008 | 0.572 ± 0.026 | 0.526 ± 0.012 |
>
> | Model | GPT-2 Surprisal | Head Word Position | Part of Speech | Word Length |
> |-------|---------|---------|---------|---------|
> | Linear (voltage) | 0.529 ± 0.006 | 0.537 ± 0.005 | 0.530 ± 0.005 | 0.536 ± 0.008 |
> | Linear (spectrogram) | 0.535 ± 0.006 | 0.557 ± 0.011 | 0.531 ± 0.007 | 0.537 ± 0.009 |
> | Linear (Laplacian+spectrogram) | 0.547 ± 0.007 | **0.580 ± 0.009** | **0.549 ± 0.007** | **0.571 ± 0.007** |
> | BrainBERT (untrained, frozen) | 0.534 ± 0.008 | 0.579 ± 0.015 | 0.515 ± 0.005 | 0.541 ± 0.010 |
> | BrainBERT (frozen) | 0.534 ± 0.008 | 0.577 ± 0.015 | 0.517 ± 0.006 | 0.540 ± 0.009 |
> | PopulationTransformer | **0.558 ± 0.014** | 0.523 ± 0.006 | 0.505 ± 0.006 | 0.526 ± 0.006 |
>
> | Model | Global Optical Flow | Local Optical Flow | Frame Brightness | Number of Faces |
> |-------|---------|---------|---------|---------|
> | Linear (voltage) | 0.510 ± 0.002 | 0.506 ± 0.002 | 0.504 ± 0.004 | 0.503 ± 0.002 |
> | Linear (spectrogram) | 0.538 ± 0.006 | 0.534 ± 0.008 | **0.523 ± 0.007** | 0.512 ± 0.003 |
> | Linear (Laplacian+spectrogram) | **0.549 ± 0.006** | **0.539 ± 0.006** | 0.514 ± 0.008 | **0.516 ± 0.006** |
> | BrainBERT (untrained, frozen) | 0.509 ± 0.004 | 0.508 ± 0.003 | 0.499 ± 0.002 | 0.497 ± 0.003 |
> | BrainBERT (frozen) | 0.510 ± 0.003 | 0.512 ± 0.003 | 0.506 ± 0.006 | 0.499 ± 0.003 |
> | PopulationTransformer | 0.518 ± 0.010 | 0.516 ± 0.012 | 0.509 ± 0.013 | 0.497 ± 0.008 |

---

> > ### Author Response · Authors · 2025-12-03
> > **Supporting tables (3/3)**
> >
> > # Performance Comparison - WithinSession (Neuroprobe MultiClass)
> >
> > Performance comparison across tasks (mean ± SEM). **Best performing model** for each task is shown in bold.
> >
> > | Model | Overall | Sentence Onset | Speech | Volume |
> > |-------|---------|---------|---------|---------|
> > | Linear (voltage) | 0.575 ± 0.003 | 0.795 ± 0.021 | 0.656 ± 0.022 | 0.539 ± 0.008 |
> > | Linear (spectrogram) | 0.593 ± 0.004 | 0.851 ± 0.025 | 0.825 ± 0.028 | **0.615 ± 0.022** |
> > | Linear (Laplacian+spectrogram) | **0.617 ± 0.003** | **0.891 ± 0.018** | **0.883 ± 0.018** | 0.612 ± 0.019 |
> > | BrainBERT (untrained, frozen) | 0.560 ± 0.003 | 0.752 ± 0.029 | 0.598 ± 0.021 | 0.529 ± 0.006 |
> > | BrainBERT (frozen) | 0.560 ± 0.003 | 0.753 ± 0.029 | 0.606 ± 0.022 | 0.530 ± 0.006 |
> > | PopulationTransformer | 0.546 ± 0.005 | 0.725 ± 0.043 | 0.687 ± 0.034 | 0.561 ± 0.016 |
> >
> > | Model | Delta Volume | Voice Pitch | Word Position | Inter-word Gap |
> > |-------|---------|---------|---------|---------|
> > | Linear (voltage) | 0.623 ± 0.011 | 0.520 ± 0.003 | 0.696 ± 0.016 | 0.546 ± 0.007 |
> > | Linear (spectrogram) | 0.609 ± 0.017 | 0.532 ± 0.005 | 0.643 ± 0.023 | 0.542 ± 0.012 |
> > | Linear (Laplacian+spectrogram) | **0.640 ± 0.017** | **0.541 ± 0.007** | **0.712 ± 0.022** | **0.558 ± 0.008** |
> > | BrainBERT (untrained, frozen) | 0.599 ± 0.012 | 0.516 ± 0.005 | 0.640 ± 0.021 | 0.547 ± 0.010 |
> > | BrainBERT (frozen) | 0.599 ± 0.013 | 0.513 ± 0.005 | 0.641 ± 0.022 | 0.543 ± 0.012 |
> > | PopulationTransformer | 0.602 ± 0.027 | 0.508 ± 0.011 | 0.542 ± 0.020 | 0.517 ± 0.014 |
> >
> > | Model | GPT-2 Surprisal | Head Word Position | Part of Speech | Word Length |
> > |-------|---------|---------|---------|---------|
> > | Linear (voltage) | 0.539 ± 0.006 | 0.570 ± 0.008 | 0.537 ± 0.005 | 0.562 ± 0.007 |
> > | Linear (spectrogram) | 0.526 ± 0.009 | 0.565 ± 0.012 | 0.528 ± 0.008 | 0.538 ± 0.010 |
> > | Linear (Laplacian+spectrogram) | **0.555 ± 0.008** | **0.602 ± 0.012** | **0.551 ± 0.008** | **0.569 ± 0.010** |
> > | BrainBERT (untrained, frozen) | 0.536 ± 0.007 | 0.581 ± 0.011 | 0.524 ± 0.004 | 0.547 ± 0.010 |
> > | BrainBERT (frozen) | 0.537 ± 0.008 | 0.583 ± 0.012 | 0.522 ± 0.004 | 0.549 ± 0.011 |
> > | PopulationTransformer | 0.516 ± 0.011 | 0.511 ± 0.004 | 0.499 ± 0.008 | 0.507 ± 0.009 |
> >
> > | Model | Global Optical Flow | Local Optical Flow | Frame Brightness | Number of Faces |
> > |-------|---------|---------|---------|---------|
> > | Linear (voltage) | 0.526 ± 0.006 | 0.521 ± 0.003 | 0.500 ± 0.008 | 0.498 ± 0.005 |
> > | Linear (spectrogram) | 0.552 ± 0.008 | 0.539 ± 0.012 | **0.518 ± 0.010** | 0.507 ± 0.004 |
> > | Linear (Laplacian+spectrogram) | **0.564 ± 0.009** | **0.552 ± 0.010** | 0.513 ± 0.013 | **0.514 ± 0.007** |
> > | BrainBERT (untrained, frozen) | 0.514 ± 0.005 | 0.514 ± 0.004 | 0.503 ± 0.003 | 0.505 ± 0.004 |
> > | BrainBERT (frozen) | 0.511 ± 0.005 | 0.513 ± 0.006 | 0.501 ± 0.005 | 0.503 ± 0.004 |
> > | PopulationTransformer | 0.519 ± 0.010 | 0.516 ± 0.010 | 0.482 ± 0.016 | 0.498 ± 0.010 |

---

### Meta-Review · Area_Chair_M2kW · 2026-01-02

**Summary:**

The paper presents a well-considered and carefully implemented evaluation work with clear potential value. However, it remains unclear how the proposed evaluation will meaningfully impact the broader community benchmarks or effectively guide future discoveries. In particular, the work does not offer a unified framework or end-to-end platform that would allow researchers from different domains to easily perform decoding tasks, nor does it provide sufficiently clear open-source guidelines to support practical adoption. Moreover, several fundamental concerns remain unresolved after rebuttal, including the heavy reliance on binary task formulations and the lack of comprehensive multi-class and stability analyses. Questions regarding the representativeness of benchmark subsets and the fairness and interpretability of baseline comparisons are also only partially addressed.

**Reviewer Concerns:**

The rebuttal addresses some technical clarifications and adds limited additional analyses, improving transparency of the implementation. However, the claimed benchmark itself is not quite clear relative to evaluation without a unified framework, and the core concerns remain unresolved, including reliance on binary task formulations, lack of convincing multi-class and stability results, unclear community impact and guidance for discovery, and only partial resolution of issues around benchmark representativeness and baseline fairness.

**Reviewer Scores:**

Reviewers with initially negative scores would likely remain unchanged, given that the core concerns were not directly addressed.

---

### Decision · Program_Chairs · 2026-01-26

Reject